

# Exceptional points and pseudo-Hermiticity in real potential scattering

Farhang Loran[1*] and Ali Mostafazadeh[2†]

**1** Department of Physics, Isfahan University of Technology, Isfahan 84156-83111, Iran
**2** Departments of Mathematics and Physics, Koç University, 34450 Sarıyer, Istanbul, Turkey

⋆ loran@iut.ac.ir, † amostafazadeh@ku.edu.tr

## Abstract

We employ a recently-developed transfer-matrix formulation of scattering theory in two dimensions to study a class of scattering setups modeled by real potentials. The transfer matrix for these potentials is related to the time-evolution operator for an associated pseudo-Hermitian Hamiltonian operator $\widehat{\mathbf{H}}$ which develops an exceptional point for a discrete set of incident wavenumbers. We use the spectral properties of this operator to determine the transfer matrix of these potentials and solve their scattering problem. We apply our general results to explore the scattering of waves by a waveguide of finite length in two dimensions, where the source of the incident wave and the detectors measuring the scattered wave are positioned at spatial infinities while the interior of the waveguide, which is filled with an inactive material, forms a finite rectangular region of the space. The study of this model allows us to elucidate the physical meaning and implications of the presence of the real and complex eigenvalues of $\widehat{\mathbf{H}}$ and its exceptional points. Our results reveal the relevance of the concepts of pseudo-Hermitian operator and exceptional point in the standard quantum mechanics of closed systems where the potentials are required to be real.



# 1   Introduction

The term exceptional point usually refers to a point $R_\star$ in the space of parameters $R$ of a linear operator $H[R]$ such that every perturbation of $R_\star$ changes the number of linearly-independent eigenvectors corresponding to one or more of the eigenvalues of $H[R_\star]$, [1]. In other words, it is a point where two or more of the eigenvectors for the same eigenvalue coalesce. The fact that this can never happen for a Hermitian operator has naturally led to the belief that exceptional points do not play any role in quantum mechanics of closed systems where the observables and the Hamiltonian operator are required to be Hermitian. In this article, we provide compelling evidence against this belief by identifying a set of exceptional points that arise in the treatment of the two-dimensional scattering problem for real potentials of the form,

$$v(x, y) = \begin{cases} \mathscr{V}(y) & \text{for} \quad x \in [a_-, a_+], \\ 0 & \text{for} \quad x \notin [a_-, a_+]. \end{cases} \tag{1}$$

Here $a_\pm$ are real parameters such that $a_- < a_+$, and $\mathscr{V}(y)$ is a real-valued function that tends to infinity as $y \to \pm\infty$.

Exceptional points entered physics literature through the works of physicists interested in effective non-Hermitian matrix Hamiltonians describing open quantum systems [2–7] and the geometric phases induced by their time-dependent variants [8–11]. See also [12]. The discovery of the interesting spectral properties of the Schrödinger operator, $-\partial_x^2 + v(x)$, for complex $\mathcal{PT}$-symmetric potentials [13, 14] and the fact that the presence of exceptional points is a generic feature of these operators have boosted the interest in their study [15–17]. The developments leading to the optical realizations of $\mathcal{PT}$-symmetric potentials [18, 19] have subsequently intensified the search for applications of exceptional points in classical optics and made the subject into a fruitful area of research in both theoretical and applied physics [20–27]. The present investigation differs from the earlier works on the physical aspects of exceptional points in that it deals with exceptional points arising in the treatment of a scattering problem for a real potential. In the context of their optical or acoustic realizations, these are exceptional points whose presence does not require active or lossy materials.

Standard approaches to potential scattering rely on general assumptions on the asymptotic decay rate of the potential. In particular both the short- and long-range potentials considered in the mathematical theories of scattering require the potential to tend to zero at spatial infinities [28–30]. There are, however, physical situations where one needs to deal with the scattering by an interaction that has a nonzero strength in an infinitely extended region of space. A typical example is the scattering problem for a grating potential of the form (1) with $\mathscr{V}(y)$ being a periodic function. Because this class of potentials have a finite range along the $x$-axis, it makes sense to speak of the asymptotic plane-wave solutions of the corresponding Schrödinger equation,

$$\left[ -\partial_x^2 - \partial_y^2 + v(x, y) \right] \psi(x, y) = k^2 \psi(x, y), \tag{2}$$

where $k$ is the incident wavenumber and "asymptotic" refers to the limits $x \to \pm\infty$. In particular, we can employ the standard definition of the scattering solutions of (2) to formulate the scattering problem for such a potential, except in the vicinity of the $y$-axis. These are solutions satisfying,

$$\psi(\mathbf{r}) \to \frac{1}{2\pi} \left[ e^{i\mathbf{k}_0 \cdot \mathbf{r}} + \sqrt{\frac{i}{kr}} \, e^{ikr} \mathfrak{f}(\theta) \right] \quad \text{for} \ \ r \to \infty \ \ \text{and} \ \ \theta \neq \pm\frac{\pi}{2}, \tag{3}$$

where $\mathbf{r}$ is the position vector having $(x, y)$ as its Cartesian coordinates, $\mathbf{k}_0$ is the incident wave vector, $(r, \theta)$ are the polar coordinates of $\mathbf{r}$, and $\mathfrak{f}$ is the scattering amplitude [31].

Recently, we have developed a transfer-matrix formulation of potential scattering in two and three dimensions that is suitable for exploring the scattering phenomenon defined by potentials having a short range along the scattering axis[1] [32,33]. In the present article, we employ this formulation to explore the scattering properties of the potentials (1). This requires the computation of the corresponding transfer matrix which is a linear operator acting in an infinite-dimensional function space. We achieve this by constructing the spectral resolution of a related pseudo-Hermitian operator $\widehat{\mathbf{H}}$, [34]. This operator develops exceptional points at a discrete set of wavenumbers. Our main purpose is to explore the effects of these exceptional points on the scattering properties of the potential. To achieve this we offer a comprehensive treatment of the scattering problem for a waveguide having a finite length.

The organization of this article is as follows. In Sec. 2, we review the transfer-matrix formulation of stationary scattering in two dimensions and determine $\widehat{\mathbf{H}}$ for the potentials of the form (1). Here we introduce the two-dimensional analogs of the reflection and transmission amplitudes of potential scattering in one dimension and provide a representation of the S-matrix which resembles its one-dimensional analog's. In Sec. 3, we solve the eigenvalue problem for $\widehat{\mathbf{H}}$, identify its exceptional points, and use its spectral resolution to obtain an explicit expression for the transfer matrix. In Sec. 4 we give the solution of the scattering problem for these potentials and compute their S-matrix. In Sec. 5 we address the scattering problem for a finite-size waveguide. Here we explore the physical meaning of the solution we find for the scattering problem and discuss the implications of the presence of an exceptional point. In Sec. 6, we provide a summary of our findings and present our concluding remarks.

## 2 Transfer and scattering matrices in two dimensions

Consider a potential $v(x, y)$ that vanishes outside the region bounded by a pair of lines parallel to the $y$-axis, i.e., there are real numbers $a_\pm$ with $a_- < a_+$ such that

$$v(x, y) = 0 \ \text{ for } \ x \notin [a_-, a_+]. \tag{4}$$

Then, every bounded solution $\psi$ of the Schrödinger equation (2) satisfies

$$\psi(x, y) = \begin{cases} \displaystyle\int_{-\infty}^{\infty} \frac{dp}{4\pi^2 \varpi(p)} \left[ A_-(p) e^{i\varpi(p)x} + \mathscr{B}_-(p)e^{-i\varpi(p)x} \right] e^{ipy} & \text{for} \quad x \leq a_-, \\ \displaystyle\int_{-\infty}^{\infty} \frac{dp}{4\pi^2 \varpi(p)} \left[ \mathscr{A}_+(p) e^{i\varpi(p)x} + B_+(p)e^{-i\varpi(p)x} \right] e^{ipy} & \text{for} \quad x \geq a_+, \end{cases} \tag{5}$$

where

$$\varpi(p) := \begin{cases} \sqrt{k^2 - p^2} & \text{for} \quad |p| < k, \\ i\sqrt{p^2 - k^2} & \text{for} \quad |p| \geq k, \end{cases} \tag{6}$$

and $A_-$, $\mathscr{B}_-$, $\mathscr{A}_+$, and $B_+$ are complex-valued functions[2] such that

$$A_-(p) = B_+(p) = 0 \ \text{ for } \ |p| \geq k. \tag{7}$$

---

[1]In a two (respectively three) dimensional scattering setup, the source of the incident wave lies on a line (respectively plane) whose distance from the interaction region, where the potential has sizable strength, is large enough so that the incident wave may be approximated by a plane wave. The term "scattering axis" refers to a normal axis to this line (respectively plane) that passes through the interaction region.

[2]Ref. [33] uses the symbols $\breve{A}_-$, $\breve{\mathscr{B}}_-$, $\breve{\mathscr{A}}_+$, and $\breve{B}_+$ for what we call $A_-$, $\mathscr{B}_-$, $\mathscr{A}_+$, and $B_+$, respectively.



Figure 1: Schematic view of the scattering setup for a left-incident wave (on the left) and a right-incident wave (on the right). $\mathbf{k}_0$ and $\mathbf{k}$ are respectively the incident and scattered wave vectors. For the left- and right-incident waves, the incidence angle $\theta_0$ takes values in $(-\frac{\pi}{2}, \frac{\pi}{2})$ and $(\frac{\pi}{2}, \frac{3\pi}{2})$, respectively. The support of the potential lies between the lines $x = a_-$ and $x = a_+$ (region colored in purple.)

This relation together with (5) and (6) imply

$$\psi(x, y) \to \int_{-k}^{k} \frac{dp}{4\pi^2 \varpi(p)} \left[ A_\pm(p) e^{i\varpi(p)x} + B_\pm(p) e^{-i\varpi(p)x} \right] e^{ipy} \quad \text{for } x \to \pm\infty, \tag{8}$$

where

$$A_+ := \widehat{\Pi}_k \mathcal{A}_+, \qquad\qquad B_- := \widehat{\Pi}_k \mathcal{B}_-, \tag{9}$$

and $\widehat{\Pi}_k$ is the projection operator defined on the set $\mathscr{F}$ of complex-valued (generalized) functions of $p$ according to

$$(\widehat{\Pi}_k \phi)(p) := \begin{cases} \phi(p) & \text{for } |p| < k, \\ 0 & \text{for } |p| \geq k. \end{cases} \tag{10}$$

Introducing

$$\mathscr{F}_k := \{ \phi \in \mathscr{F} \mid \phi(p) = 0 \text{ for } |p| \geq k \},$$

we can express (7) and (9) as $A_\pm, B_\pm \in \mathscr{F}_k$.

In analogy with one dimension [35], we identify the transfer matrix $\widehat{\mathbf{M}}$ and the scattering matrix $\widehat{\mathbf{S}}$ of the potential with a pair of $2 \times 2$ matrices with operator entries that satisfy [33]

$$\widehat{\mathbf{M}} \begin{bmatrix} A_- \\ B_- \end{bmatrix} = \begin{bmatrix} A_+ \\ B_+ \end{bmatrix}, \qquad\qquad \widehat{\mathbf{S}} \begin{bmatrix} A_- \\ B_+ \end{bmatrix} = \begin{bmatrix} A_+ \\ B_- \end{bmatrix}. \tag{11}$$

Notice that these are not numerical matrices; they are linear operators acting in the infinite-dimensional function space of two-component wave functions,

$$\mathscr{F}_k^{2\times 1} := \left\{ \begin{bmatrix} \phi_+ \\ \phi_- \end{bmatrix} \,\middle|\, \phi_\pm \in \mathscr{F}_k \right\}.$$

The scattering setup for a potential fulfilling (4) involves a source of the incident wave that is located at either $x = -\infty$ or $x = +\infty$. These respectively correspond to the scattering of left- and right-incident waves where the incidence angle $\theta_0$ ranges over $(-\frac{\pi}{2}, \frac{\pi}{2})$ and $(\frac{\pi}{2}, \frac{3\pi}{2})$. See Fig. 1. In the following we use the superscript $l/r$ to label the scattering amplitude $\mathfrak{f}$ and the coefficient functions $A_\pm$ and $B_\pm$ for the left/right-incident waves.

Comparing the asymptotic expressions (3) and (8) for the left- and right-incident waves and using a result derived in [32, Appendix A], we can show that [33]

$$B_+^l(p) = A_-^r(p) = 0, \qquad A_-^l(p) = B_+^r(p) = 2\pi\varpi(p_0)\,\delta(p-p_0), \tag{12}$$

$$\mathfrak{f}^l(\theta) = \begin{cases} T^l(\theta) + i\sqrt{2\pi}\,\delta(\theta - \theta_0) & \text{for} \quad \theta \in (-\tfrac{\pi}{2}, \tfrac{\pi}{2}), \\ R^l(\theta) & \text{for} \quad \theta \in (\tfrac{\pi}{2}, \tfrac{3\pi}{2}), \end{cases} \tag{13}$$

$$\mathfrak{f}^r(\theta) = \begin{cases} R^r(\theta) & \text{for} \quad \theta \in (-\tfrac{\pi}{2}, \tfrac{\pi}{2}), \\ T^r(\theta) + i\sqrt{2\pi}\delta(\theta - \theta_0) & \text{for} \quad \theta \in (\tfrac{\pi}{2}, \tfrac{3\pi}{2}), \end{cases} \tag{14}$$

where $p_0$ is the $y$-component of the incident wave vector $\mathbf{k}_0$, i.e., $p_0 := k\sin\theta_0$, and

$$R^l(\theta) := -\frac{i}{\sqrt{2\pi}}\,B_-^l(k\sin\theta), \qquad\qquad T^l(\theta) := -\frac{i}{\sqrt{2\pi}}\,A_+^l(k\sin\theta), \tag{15}$$

$$R^r(\theta) := -\frac{i}{\sqrt{2\pi}}\,A_+^r(k\sin\theta), \qquad\qquad T^r(\theta) := -\frac{i}{\sqrt{2\pi}}\,B_-^r(k\sin\theta). \tag{16}$$

Next, we substitute (12) in (11) to establish

$$\widehat{M}_{22}B_-^l = -2\pi\varpi(p_0)\widehat{M}_{21}\,\delta_{p_0}, \qquad\qquad \widehat{M}_{22}B_-^r = 2\pi\varpi(p_0)\,\delta_{p_0}, \tag{17}$$

$$A_+^l = 2\pi\varpi(p_0)\widehat{M}_{11}\,\delta_{p_0} + \widehat{M}_{12}B_-^l, \qquad\qquad A_+^r = \widehat{M}_{12}B_-^r, \tag{18}$$

and

$$\begin{bmatrix} A_+^l & A_+^r \\ B_-^l & B_-^r \end{bmatrix} = 2\pi\varpi(p_0)\widehat{\mathbf{S}}\,\delta_{p_0}, \tag{19}$$

where $\widehat{\mathbf{I}} := \widehat{I}\,\mathbf{I}$, $\widehat{I}$ is the identity operator acting in $\mathscr{F}$, $\mathbf{I}$ is the $2\times 2$ identity matrix, and $\delta_{p_0}$ is the Dirac delta function centered at $p_0$, i.e., $\delta_{p_0}(p) := \delta(p-p_0)$.

According to (13) – (16) and (19), the knowledge of the scattering matrix $\widehat{\mathbf{S}}$ is sufficient for the determination of the scattering amplitudes and consequently the reflection and transmission amplitudes. Indeed, we can use (15), (16), and (19) to show that

$$\begin{bmatrix} T^l(\theta) & R^r(\theta) \\ R^l(\theta) & T^r(\theta) \end{bmatrix} = -i\sqrt{2\pi}\,\widehat{\mathbf{S}}\,\delta(\theta - \theta_0), \tag{20}$$

where

$$\widehat{\mathbf{S}}\,\delta(\theta - \theta_0) := k|\cos\theta_0|\big(\widehat{\mathbf{S}}\,\delta_{p_0}\big)(p) = k|\cos\theta_0|\langle p|\widehat{\mathbf{S}}|p_0\rangle \quad \text{for} \quad p = k\sin\theta. \tag{21}$$

Equation (20) is the two-dimensional analog of the well-known relation between the S-matrix and the reflection and transmission amplitudes in one dimension [36].

The transfer matrix $\widehat{\mathbf{M}}$ also contains the information about the scattering amplitudes. To compute the latter we can solve (17) for $B_-^{l/r}$, substitute it in (18) to determine $A_+^{l/r}$, and use the result in (13) – (16) to find $\mathfrak{f}^{l/r}$. This seems to make $\widehat{\mathbf{M}}$ practically less advantageous than $\widehat{\mathbf{S}}$, but it has two useful properties [33]:

1. It enjoys a composition property that is similar to that of the transfer matrix in one dimension.

2. It can be expressed in terms of the evolution operator for a certain non-unitary quantum system. In particular, it admits a Dyson series expansion.

A proper derivation of these properties requires the use of an auxiliary transfer matrix [33]. This is a linear operator $\widehat{\mathcal{M}}$ acting in

$$\mathscr{F}^{2\times 1} := \left\{ \left[ \begin{array}{c} \xi_+ \\ \xi_- \end{array} \right] \,\middle|\, \xi_\pm \in \mathscr{F} \right\},$$

that satisfies

$$\widehat{\mathcal{M}} \left[ \begin{array}{c} A_- \\ \mathscr{B}_- \end{array} \right] = \left[ \begin{array}{c} \mathscr{A}_+ \\ B_+ \end{array} \right]. \tag{22}$$

The auxiliary transfer matrix has two important features. Firstly, it is related to the (fundamental) transfer matrix $\widehat{\mathbf{M}}$ via

$$\widehat{\mathbf{M}} = \widehat{\mathbf{\Pi}}_k \, \widehat{\mathcal{M}} \, \widehat{\mathbf{\Pi}}_k, \tag{23}$$

where $\widehat{\mathbf{\Pi}}_k$ is the projection operator defined on $\mathscr{F}^{2\times 1}$ according to

$$\widehat{\mathbf{\Pi}}_k \left[ \begin{array}{c} \xi_+ \\ \xi_- \end{array} \right] := \left[ \begin{array}{c} \widehat{\Pi}_k \xi_+ \\ \widehat{\Pi}_k \xi_- \end{array} \right] \quad \text{for} \quad \left[ \begin{array}{c} \xi_+ \\ \xi_- \end{array} \right] \in \mathscr{F}^{2\times 1}, \tag{24}$$

and $\widehat{\Pi}_k$ is the projection operator given by (10). Secondly, we can express it in terms of the evolution operator $\widehat{\mathcal{U}}(x, x_0)$ for an effective non-unitary quantum system with the Hamiltonian operator,

$$\widehat{\mathcal{H}}(x) := \frac{1}{2} e^{-i\widehat{\varpi} x \sigma_3} v(x, \hat{y}) \mathcal{K} \, e^{i\widehat{\varpi} x \sigma_3} \widehat{\varpi}^{-1}, \tag{25}$$

where $x$ plays the role of time,

$$\widehat{\varpi} := \varpi(\hat{p}) = \int_{-\infty}^{\infty} dp \, \varpi(p) |p\rangle\langle p|, \tag{26}$$

$\hat{y}$ and $\hat{p}$ are respectively the $y$-component of the standard position and momentum operators, i.e., $(\hat{y}\phi)(p) = i\partial_p \phi(p)$ and $(\hat{p}\phi)(p) := p\phi(p)$,

$$\mathcal{K} := \left[ \begin{array}{cc} 1 & 1 \\ -1 & -1 \end{array} \right] = \sigma_3 + i\sigma_2,$$

and $\sigma_j$, with $j \in \{1,2,3\}$, denote the Pauli matrices. We can view $v(x, \hat{y})$ as the operator acting in $\mathscr{F}$ according to

$$\left( v(x, \hat{y})\phi \right)(p) := \frac{1}{2\pi} \int_{-\infty}^{\infty} dq \, \tilde{v}(x, p-q)\phi(q), \tag{27}$$

where a tilde over a function of $(x, y)$ stands for its Fourier transform with respect to $y$, i.e., $\tilde{f}(x, p) := \int_{-\infty}^{\infty} dy \, e^{-ipy} f(x, y)$.

It is easy to check that the time-independent Schrödinger equation (2) is equivalent to the "time-dependent" Schrödinger equation,

$$i\partial_x \Psi(x) = \widehat{\mathcal{H}}(x) \Psi(x), \tag{28}$$

provided that we identify $\Psi(x)$ with the element of $\mathscr{F}^{2\times 1}$ given by

$$\left( \Psi(x) \right)(p) := \pi e^{-ix\varpi(p)\sigma_3} \left[ \begin{array}{c} \varpi(p)\tilde{\psi}(x, p) - i\partial_x \tilde{\psi}(x, p) \\ \varpi(p)\tilde{\psi}(x, p) + i\partial_x \tilde{\psi}(x, p) \end{array} \right]. \tag{29}$$

An important consequence of (5) and (29) is that $\Psi(x)$ satisfies

$$\Psi(x) = \left[ \begin{array}{c} A_- \\ \mathscr{B}_- \end{array} \right] \text{ for } x \le a_-, \qquad\qquad \Psi(x) = \left[ \begin{array}{c} \mathscr{A}_+ \\ B_+ \end{array} \right] \text{ for } x \ge a_+. \qquad (30)$$

This together with (22) and $\Psi(x) = \widehat{\mathcal{U}}(x, x_0)\Psi(x_0)$ justify the identification of the auxiliary transfer matrix with $\widehat{\mathcal{U}}(a_+, a_-)$. Because $\widehat{\mathcal{H}}(x)$ vanishes for $x \notin [a_-, a_+]$, this is equal to $\widehat{\mathcal{U}}(x_+, x_-)$ for $x_- \le a_-$ and $x_+ \ge a_+$. In particular,

$$\widehat{\mathcal{M}} = \widehat{\mathcal{U}}(a_+, a_-) = \lim_{x_\pm \to \pm\infty} \widehat{\mathcal{U}}(x_+, x_-). \qquad (31)$$

This relation is responsible for the composition property of $\widehat{\mathcal{M}}$ and consequently of $\widehat{\mathbf{M}}$, [33].

Because the Hamiltonian operator $\widehat{\mathcal{H}}(x)$ depends on the 'time' variable $x$, the calculation of its evolution operator is generally intractable. For the potentials of the form (1), $v(x, \hat{y}) = \mathscr{V}(\hat{y})$ for $x \in [a_-, a_+]$, and we can determine the evolution operator for $\widehat{\mathcal{H}}(x)$ without much difficulty. To see this, we make the transformation,

$$\Psi(x) \to \Phi(x) := e^{ix\widehat{\varpi}\boldsymbol{\sigma}_3}\Psi(x), \qquad (32)$$

and check that, for $x \in [a_-, a_+]$, $\Phi(x)$ satisfies $i\partial_x \Phi(x) = \widehat{\mathbf{H}}\Phi(x)$ for

$$\widehat{\mathbf{H}} := \frac{1}{2}\widehat{\mathscr{V}}\widehat{\varpi}^{-1}\mathcal{K} - \widehat{\varpi}\boldsymbol{\sigma}_3, \qquad (33)$$

where $\widehat{\mathscr{V}} := \mathscr{V}(\hat{y})$. The fact that $\widehat{\mathbf{H}}$ is $x$-independent allows us to express its evolution operator in the form $\widehat{\mathbf{U}}(x, x_0) = e^{-i(x-x_0)\widehat{\mathbf{H}}}$. In view of (32), $\widehat{\mathcal{U}}(x, x_0) = e^{-ix\widehat{\varpi}\boldsymbol{\sigma}_3}e^{-i(x-x_0)\widehat{\mathbf{H}}}e^{ix_0\widehat{\varpi}\boldsymbol{\sigma}_3}$, whenever $x$ and $x_0$ belong to $[a_-, a_+]$. Substituting this relation in (31) and making use of (23), we find

$$\widehat{\mathcal{M}} = \widehat{\mathcal{U}}(a_+, a_-) = e^{-ia_+\widehat{\varpi}\boldsymbol{\sigma}_3}e^{-ia\widehat{\mathbf{H}}}e^{ia_-\widehat{\varpi}\boldsymbol{\sigma}_3}, \qquad (34)$$

$$\begin{aligned} \widehat{\mathbf{M}} &= \widehat{\Pi}_k e^{-ia_+\widehat{\varpi}\boldsymbol{\sigma}_3}e^{-ia\widehat{\mathbf{H}}}e^{ia_-\widehat{\varpi}\boldsymbol{\sigma}_3}\widehat{\Pi}_k \\ &= e^{-ia_+\widehat{\varpi}\boldsymbol{\sigma}_3}\widehat{\Pi}_k e^{-ia\widehat{\mathbf{H}}}\widehat{\Pi}_k e^{ia_-\widehat{\varpi}\boldsymbol{\sigma}_3}, \end{aligned} \qquad (35)$$

where $a := a_+ - a_-$, and we have benefitted from (24) and the fact that $\widehat{\Pi}_k$ and $\hat{p}$ commute.

## 3 Determination of the transfer matrix

For potentials of the form (1), (35) reduces the calculation of the transfer matrix $\widehat{\mathbf{M}}$ to that of $e^{-ia\widehat{\mathbf{H}}}$. In this section, we perform this calculation for situations where $\mathscr{V}$ is a real confining potential, i.e., $\mathscr{V}(y) \to +\infty$ for $y \to \pm\infty$. In this case, the Schrödinger operator, $\hat{p}^2 + \widehat{\mathscr{V}}$, acts as a Hermitian operator in the Hilbert space $L^2(\mathbb{R})$ of square integrable functions of $y$ and has a real and discrete spectrum consisting of nondegenerate eigenvalues $E_n$, where $n$ ranges over the set of positive integers.

The Hamiltonian operator $\widehat{\mathbf{H}}$ is manifestly non-Hermitian. Yet its particular form allows for the solution of its eigenvalue problem. To see this, first we introduce

$$|\mathcal{X}_n\rangle := \left[ \begin{array}{c} -1 \\ 1 \end{array} \right]|\phi_n\rangle, \quad |\mathcal{Y}_n\rangle := \frac{1}{k}\left[ \begin{array}{c} 1 \\ 1 \end{array} \right]\widehat{\varpi}|\phi_n\rangle, \quad |\mathcal{Z}_n\rangle := k\left[ \begin{array}{c} 1 \\ 1 \end{array} \right]\widehat{\varpi}^{-1\dagger}|\phi_n\rangle, \qquad (36)$$

where $|\phi_n\rangle$ are the eigenvectors of $\hat{p}^2 + \widehat{\mathscr{V}}$ that form an orthonormal basis of $L^2(\mathbb{R})$; they satisfy

$$(\hat{p}^2 + \widehat{\mathscr{V}})|\phi_n\rangle = E_n|\phi_n\rangle, \qquad (37)$$

$\langle\phi_m|\phi_n\rangle = \delta_{mn}$, and $\sum_{n=1}^{\infty}|\phi_n\rangle\langle\phi_n| = \widehat{I}$, where $\widehat{I}$ is the identity operator. We can use (6) and (37) to show that

$$\left(\widehat{\varpi}^2 - \widehat{\mathcal{V}}\right)|\phi_n\rangle = w_n^2|\phi_n\rangle, \tag{38}$$

where

$$w_n := \begin{cases} \sqrt{k^2 - E_n} & \text{for} \quad E_n \le k^2, \\ i\sqrt{E_n - k^2} & \text{for} \quad E_n > k^2. \end{cases} \tag{39}$$

In view of (33), (36), and (38), it is easy to check that

$$\widehat{\mathbf{H}}|\mathcal{X}_n\rangle = k|\mathcal{Y}_n\rangle, \qquad\qquad \widehat{\mathbf{H}}|\mathcal{Y}_n\rangle = \frac{w_n^2}{k}|\mathcal{X}_n\rangle, \tag{40}$$

$$\widehat{\mathbf{H}}^{\dagger}|\mathcal{X}_n\rangle = \frac{w_n^2}{k}|\mathcal{Z}_n\rangle, \qquad\qquad \widehat{\mathbf{H}}^{\dagger}|\mathcal{Z}_n\rangle = k|\mathcal{X}_n\rangle. \tag{41}$$

These relations identify the spans of $\{|\mathcal{X}_n\rangle,|\mathcal{Y}_n\rangle\}$ and $\{|\mathcal{X}_n\rangle,|\mathcal{Z}_n\rangle\}$ with invariant subspaces of $\widehat{\mathbf{H}}$ and $\widehat{\mathbf{H}}^{\dagger}$, respectively. This reduces the eigenvalue problem for these operators to that of the $2 \times 2$ matrices,

$$\begin{bmatrix} 0 & w_n^2/k \\ k & 0 \end{bmatrix},$$

and leads to the following observations.

1. The spectrum of $\widehat{\mathbf{H}}$ (and $\widehat{\mathbf{H}}^{\dagger}$) consists of eigenvalues of the form $\pm w_n$. Because $w_n$ are either real or imaginary, it is invariant under reflections about both the real and imaginary axes in the complex plane.

2. $\widehat{\mathbf{H}}$ has finitely many (or no) real eigenvalues and infinitely many complex-conjugate pairs of eigenvalues. The number of its real eigenvalues depends on the wavenumber $k$.

3. For $k = \sqrt{E_n}$, $w_n$ vanishes, and $\widehat{\mathbf{H}}$ becomes non-diagonalizable. In particular, these values of the wavenumber mark the exceptional points of $\widehat{\mathbf{H}}$.

4. As one increases $k$, complex-conjugate pairs of eigenvalues merge, become zero at exceptional points, and turn into pairs of real eigenvalues with opposite sign.

In view of the characterization theorems given in Refs. [34, 37–39], these show that $\widehat{\mathbf{H}}$ is a pseudo-Hermitian operator. The same holds for $i\widehat{\mathbf{H}}$.[3]

We can use (40) and (41) to construct a biorthonormal system consisting of the (generalized) eigenvectors of $\widehat{\mathbf{H}}$ and $\widehat{\mathbf{H}}^{\dagger}$, [41]. To do this, first we consider situations where $k^2$ does not belong to the spectrum of $\hat{p}^2 + \widehat{\mathcal{V}}$, so that $w_n \neq 0$ for all $n \in \mathbb{Z}^+$, and $\widehat{\mathbf{H}}$ is diagonalizable. Let

$$\widehat{W} := \sum_{n=1}^{\infty} w_n|\phi_n\rangle\langle\phi_n| \tag{42}$$

$$= \sqrt{k^2\widehat{I} - (\hat{p}^2 + \widehat{\mathcal{V}})} = \sqrt{\widehat{\varpi}^2 - \widehat{\mathcal{V}}}. \tag{43}$$

Then it is not difficult to show that the two-component wave functions defined by

$$|\Psi_{n,\pm}\rangle := \frac{1}{2k}\begin{bmatrix} \widehat{\varpi} \mp \widehat{W} \\ \widehat{\varpi} \pm \widehat{W} \end{bmatrix}|\phi_n\rangle, \qquad \Phi_{n,\pm} := \frac{k}{2}\begin{bmatrix} \widehat{\varpi}^{-1\dagger} \mp \widehat{W}^{\dagger-1} \\ \widehat{\varpi}^{-1\dagger} \pm \widehat{W}^{\dagger-1} \end{bmatrix}|\phi_n\rangle, \tag{44}$$

---

[3]It is also easy to check that $\{\boldsymbol{\sigma}_1, \widehat{\mathbf{H}}\} = \mathbf{0}$. In the terminology of Ref [40], this signifies a "chiral symmetry" of $\widehat{\mathbf{H}}$ which we can interpret as the reason for its eigenvalues coming in pairs of opposite sign. We can also use the pseudo-Hermiticity of $\widehat{\mathbf{H}}$ and $i\widehat{\mathbf{H}}$ to infer the existence of antilinear involutions commuting with these operator [38], i.e., there are antilinear operators $\widehat{\mathfrak{S}}$ and $\widehat{\chi}$ such that $[\widehat{\mathbf{H}}, \widehat{\mathfrak{S}}] = \widehat{\mathbf{0}}$, $\{\widehat{\mathbf{H}}, \widehat{\chi}\} = \widehat{\mathbf{0}}$, and $\widehat{\mathfrak{S}}^2 = \widehat{\chi}^2 = \widehat{\mathbf{I}}$.

satisfy

$$\widehat{\mathbf{H}}|\mathbf{\Psi}_{n,\pm}\rangle = \pm w_n|\mathbf{\Psi}_{n,\pm}\rangle\,, \qquad\qquad \widehat{\mathbf{H}}^\dagger|\mathbf{\Phi}_{n,\pm}\rangle = \pm w_n^*|\mathbf{\Phi}_{n,\pm}\rangle\,, \qquad (45)$$

$$\langle\mathbf{\Phi}_{m,\mu}|\mathbf{\Psi}_{n,\nu}\rangle = \delta_{mn}\delta_{\mu\nu}\,, \qquad\qquad \sum_{n=1}^\infty\Big(|\mathbf{\Psi}_{n,+}\rangle\langle\mathbf{\Phi}_{n,+}| + |\mathbf{\Psi}_{n,-}\rangle\langle\mathbf{\Phi}_{n,-}|\Big) = \widehat{\mathbf{I}}\,, \qquad (46)$$

where $\widehat{\mathbf{I}}$ is the identity operator acting in the Hilbert space $\mathscr{H} := \mathbb{C}^2\otimes L^2(\mathbb{R})$ of two-component square-integrable wave functions, $\langle\cdot|\cdot\rangle$ is the standard $L^2$-inner product on this space, $m, n\in\mathbb{Z}^+$, and $\mu, \nu\in\{-,+\}$.

Equations (45) and (46) show that whenever $k^2\neq E_n$ for all $n\in\mathbb{Z}^+$, $\{\mathbf{\Psi}_{n,\pm}, \mathbf{\Phi}_{n,\pm}\}$ forms a complete biorthonormal system of eigenvectors of $\widehat{\mathbf{H}}$ and $\widehat{\mathbf{H}}^\dagger$ for the Hilbert space $\mathscr{H}$. This leads to the following spectral resolutions of $\widehat{\mathbf{H}}$ and $e^{-ix\widehat{\mathbf{H}}}$.

$$\widehat{\mathbf{H}} = \sum_{n=1}^\infty w_n\Big(|\mathbf{\Psi}_{n,+}\rangle\langle\mathbf{\Phi}_{n,+}| - |\mathbf{\Psi}_{n,-}\rangle\langle\mathbf{\Phi}_{n,-}|\Big)\,, \qquad (47)$$

$$e^{-ix\widehat{\mathbf{H}}} = \sum_{n=1}^\infty\Big(e^{-iw_n x}|\mathbf{\Psi}_{n,+}\rangle\langle\mathbf{\Phi}_{n,+}| + e^{iw_n x}|\mathbf{\Psi}_{n,-}\rangle\langle\mathbf{\Phi}_{n,-}|\Big)\,. \qquad (48)$$

Substituting (44) in (48) and simplifying the resulting equation, we find

$$e^{-ix\widehat{\mathbf{H}}} = \frac{1}{2}\Big\{\widehat{\varpi}\widehat{C}(x)\widehat{\varpi}^{-1}(\mathbf{I}+\boldsymbol{\sigma}_1) + \widehat{C}(x)(\mathbf{I}-\boldsymbol{\sigma}_1) + i\big[\widehat{W}^2\widehat{S}(x)\widehat{\varpi}^{-1}\mathcal{K} + \widehat{\varpi}\widehat{S}(x)\mathcal{K}^T\big]\Big\}\,, \qquad (49)$$

where

$$\widehat{C}(x) \quad := \quad \sum_{n=1}^\infty\cos(w_n x)|\phi_n\rangle\langle\phi_n| = \cos(x\widehat{W})\,, \qquad (50)$$

$$\widehat{S}(x) \quad := \quad \sum_{n=1}^\infty w_n^{-1}\sin(w_n x)|\phi_n\rangle\langle\phi_n| = \widehat{W}^{-1}\sin(x\widehat{W})\,, \qquad (51)$$

$\widehat{W}$ is the operator defined in (42), and $\mathcal{K}^T$ stands for the transpose of $\mathcal{K}$. Notice that both $\widehat{C}(x)$ and $\widehat{S}(x)$ are even functions of $\widehat{W}$; in view of (43), they are analytic functions of $k^2\widehat{I} - (\hat{p}^2 + \widehat{\mathcal{V}}) = \widehat{\varpi}^2 - \widehat{\mathcal{V}}$.

Having determined $e^{-ix\widehat{\mathbf{H}}}$, we can use (35) to obtain the following more explicit expression for the transfer matrix of the potential (1).

$$\widehat{\mathbf{M}} = \frac{1}{2}e^{-ia_+\widehat{\varpi}\boldsymbol{\sigma}_3}\Big[\widehat{\varpi}\widehat{\mathscr{C}}\widehat{\varpi}^{-1}(\mathbf{I}+\boldsymbol{\sigma}_1) + \widehat{\mathscr{C}}(\mathbf{I}-\boldsymbol{\sigma}_1) + i\big(\widehat{\mathscr{R}}\widehat{\varpi}^{-1}\mathcal{K} + \widehat{\varpi}\widehat{\mathscr{S}}\mathcal{K}^T\big)\Big]e^{ia_-\widehat{\varpi}\boldsymbol{\sigma}_3}\,, \qquad (52)$$

where

$$\widehat{\mathscr{C}} := \widehat{\Pi}_k\widehat{C}(a)\widehat{\Pi}_k\,, \qquad\qquad \widehat{\mathscr{R}} := \widehat{\Pi}_k\widehat{W}^2\widehat{S}(a)\widehat{\Pi}_k \qquad\qquad \widehat{\mathscr{S}} := \widehat{\Pi}_k\widehat{S}(a)\widehat{\Pi}_k\,. \qquad (53)$$

Next, we consider the scattering of incident waves whose wavenumber has the value $\sqrt{E_{n_\star}}$ for some positive integer $n_\star$. We call them "exceptional wavenumbers" and use $k_{n_\star}$ to label them. Then $w_{n_\star} = 0$, $\widehat{W}$ does not have an inverse, and the restriction of $\widehat{\mathbf{H}}$ to the invariant subspace $\mathscr{H}_{n_\star}$ spanned by $\{\mathcal{X}_{n_\star}, \mathcal{Y}_{n_\star}\}$ and consequently $\widehat{\mathbf{H}}$ are not diagonalizable. In this case, 0 is a defective eigenvalue of $\widehat{\mathbf{H}}$, and the corresponding eigenvectors are proportional to $\mathcal{Y}_{n_\star}$. In particular, $\mathbf{\Psi}_{n_\star,\pm} = \mathcal{Y}_{n_\star}/2$. This shows that the set of the eigenvectors $\mathbf{\Psi}_{n,\pm}$ is not a basis of $\mathscr{H}$. We can however extend it to a basis by adjoining a generalized eigenvector associated with the eigenvalue 0, [42]. According to (40), $\mathcal{X}_{n_\star}$ is such a generalized eigenvector of $\widehat{\mathbf{H}}$.

Let us introduce

$$|\Psi_{n_\star}^+\rangle := |\Psi_{n_\star,+}\rangle = \frac{1}{2}\mathcal{Y}_{n_\star} = \frac{1}{2k}\begin{bmatrix} 1 \\ 1 \end{bmatrix}\widehat{\varpi}|\phi_{n_\star}\rangle, \quad |\Psi_{n_\star}^-\rangle := \frac{1}{2}|\mathcal{X}_{n_\star}\rangle = \frac{1}{2}\begin{bmatrix} -1 \\ 1 \end{bmatrix}|\phi_{n_\star}\rangle, \quad (54)$$

$$|\Phi_{n_\star}^+\rangle := |\mathcal{Z}_{n_\star}\rangle = k\begin{bmatrix} 1 \\ 1 \end{bmatrix}\widehat{\varpi}^{-1\dagger}|\phi_{n_\star}\rangle, \qquad |\Phi_{n_\star}^-\rangle := |\mathcal{X}_{n_\star}\rangle = \begin{bmatrix} -1 \\ 1 \end{bmatrix}|\phi_{n_\star}\rangle. \quad (55)$$

Then we can use (36) – (41) and (44) to show that the sets,

$$\mathcal{B} := \left\{ |\Psi_{n,\pm}\rangle \,\Big|\, n \in \mathbb{Z}^+ \setminus \{n_\star\} \right\} \cup \left\{ |\Psi_{n_\star}^+\rangle, |\Psi_{n_\star}^-\rangle \right\}, \tag{56}$$

$$\mathcal{B}_\perp := \left\{ |\Phi_{n,\pm}\rangle \,\Big|\, n \in \mathbb{Z}^+ \setminus \{n_\star\} \right\} \cup \left\{ |\Phi_{n_\star}^+\rangle, |\Phi_{n_\star}^-\rangle \right\}, \tag{57}$$

are bases of $\mathscr{H}$ that are biorthonormal dual of one another [41], i.e., for all $m, n \in \mathbb{Z}^+ \setminus \{n_\star\}$ and $\mu, \nu \in \{-, +\}$, we have

$$\langle\Phi_{m,\mu}|\Psi_{n,\nu}\rangle = \delta_{mn}\delta_{\mu\nu}, \qquad \langle\Phi_{n_\star}^\mu|\Psi_{n_\star}^\nu\rangle = \delta_{\mu\nu}, \qquad \langle\Phi_{n_\star}^\mu|\Psi_{n,\nu}\rangle = \langle\Phi_{m,\mu}|\Psi_{n_\star}^\nu\rangle = 0, \tag{58}$$

$$|\Psi_{n_\star}^+\rangle\langle\Phi_{n_\star}^+| + |\Psi_{n_\star}^-\rangle\langle\Phi_{n_\star}^-| + \sum_{\substack{n=1 \\ n \neq n_\star}}^{\infty}\Big(|\Psi_{n,+}\rangle\langle\Phi_{n,+}| + |\Psi_{n,-}\rangle\langle\Phi_{n,-}|\Big) = \widehat{\mathbf{I}}. \tag{59}$$

Furthermore, $|\Psi_{n_\star}^-\rangle$ and $|\Phi_{n_\star}^-\rangle$ are generalized eigenvectors of $\widehat{\mathbf{H}}$ and $\widehat{\mathbf{H}}^\dagger$, and $\mathcal{B} \setminus \{\Psi_{n_\star}^-\}$ and $\mathcal{B}_\perp \setminus \{\Phi_{n_\star}^-\}$ consist of their eigenvectors, respectively. These observations together with Eqs. (36) – (41), (58), and (59) justify the following spectral expansions of $\widehat{\mathbf{H}}$ and $e^{-ix\widehat{\mathbf{H}}}$ for $k = k_{n_\star} = \sqrt{E_{n_\star}}$.

$$\widehat{\mathbf{H}} = k|\Psi_{n_*}^+\rangle\langle\Phi_{n_*}^-| + \sum_{\substack{n=1 \\ n \neq n_\star}}^{\infty} w_n\Big(|\Psi_{n,+}\rangle\langle\Phi_{n,+}| - |\Psi_{n,-}\rangle\langle\Phi_{n,-}|\Big), \tag{60}$$

$$e^{-ix\widehat{\mathbf{H}}} = |\Psi_{n_\star}^+\rangle\langle\Phi_{n_\star}^+| + |\Psi_{n_\star}^-\rangle\langle\Phi_{n_\star}^-| - ikx|\Psi_{n_*}^+\rangle\langle\Phi_{n_*}^-| \tag{61}$$

$$+ \sum_{\substack{n=1 \\ n \neq n_\star}}^{\infty}\Big(e^{-iw_n x}|\Psi_{n,+}\rangle\langle\Phi_{n,+}| + e^{iw_n x}|\Psi_{n,-}\rangle\langle\Phi_{n,-}|\Big)$$

$$= \widehat{\mathbf{I}} - ikx|\Psi_{n_*}^+\rangle\langle\Phi_{n_*}^-| + \sum_{\substack{n=1 \\ n \neq n_\star}}^{\infty}\Big[(e^{-iw_n x} - 1)|\Psi_{n,+}\rangle\langle\Phi_{n,+}| + (e^{iw_n x} - 1)|\Psi_{n,-}\rangle\langle\Phi_{n,-}|\Big].$$

With the help of the identities,

$$|\Psi_{n_\star}^+\rangle\langle\Phi_{n_\star}^+| + |\Psi_{n_\star}^-\rangle\langle\Phi_{n_\star}^-| = \frac{1}{2}\Big[\widehat{\varpi}|\phi_{n_\star}\rangle\langle\phi_{n_\star}|\widehat{\varpi}^{-1}(\mathbf{I} + \boldsymbol{\sigma}_3) + |\phi_{n_\star}\rangle\langle\phi_{n_\star}|(\mathbf{I} - \boldsymbol{\sigma}_3)\Big], \tag{62}$$

$$|\Psi_{n_*}^+\rangle\langle\Phi_{n_*}^-| = -\frac{1}{2k}\widehat{\varpi}|\phi_{n_\star}\rangle\langle\phi_{n_\star}|\mathcal{K}^T, \tag{63}$$

which follow from (54) and (55), we have shown that (49) holds also for $k = k_{n_\star}$.[4] This in turn implies that the expression (52) for the transfer matrix $\widehat{\mathbf{M}}$ is valid also for the exceptional wavenumbers. Note however that this expression hides the signature of the exceptional point.

---

[4]Note that because $\widehat{S}(x) := \widehat{W}^{-1}\sin(x\widehat{W}) = x\Big[\widehat{I} + \sum_{\ell=1}^{\infty}\frac{(-1)^\ell x^{2\ell}}{(2\ell+1)!}\widehat{W}^{2\ell}\Big]$, this operator is defined also for the exceptional wavenumbers where $\widehat{W}$ does not have an inverse.

This is related to the third term on the right-hand side of (61). In view of (35), (61), and (63), it contributes to the transfer matrix $\widehat{\mathbf{M}}$ the term,

$$\frac{i}{2} a \, \widehat{\Pi}_k \widehat{\varpi} |\phi_{n_\star}\rangle\langle\phi_{n_\star}| \widehat{\Pi}_k \mathcal{K}^T , \tag{64}$$

which is linear in $a$.

## 4 Solution of the scattering problem

We can use (52) to obtain the following formulas for the entries of the transfer matrix $\widehat{\mathbf{M}}$.

$$\widehat{M}_{11} = e^{-ia_+\widehat{\varpi}}(\widehat{\mathscr{C}}_+ + \widehat{\mathscr{S}}_+)e^{ia_-\widehat{\varpi}} , \qquad \widehat{M}_{12} = e^{-ia_+\widehat{\varpi}}(\widehat{\mathscr{C}}_- + \widehat{\mathscr{S}}_-)e^{-ia_-\widehat{\varpi}} , \tag{65}$$

$$\widehat{M}_{21} = e^{ia_+\widehat{\varpi}}(\widehat{\mathscr{C}}_- - \widehat{\mathscr{S}}_-)e^{ia_-\widehat{\varpi}} , \qquad \widehat{M}_{22} = e^{ia_+\widehat{\varpi}}(\widehat{\mathscr{C}}_+ - \widehat{\mathscr{S}}_+)e^{-ia_-\widehat{\varpi}} , \tag{66}$$

where

$$\widehat{\mathscr{C}}_\pm := \widehat{\varpi}\widehat{\mathscr{C}}\widehat{\varpi}^{-1} \pm \widehat{\mathscr{C}} , \qquad \widehat{\mathscr{S}}_\pm := i(\widehat{\mathscr{R}}\widehat{\varpi}^{-1} \pm \widehat{\varpi}\widehat{\mathscr{S}}) . \tag{67}$$

Writing Eqs. (17) and (18) in the form

$$B_-^l = -2\pi\varpi(p_0)\widehat{M}_{22}^{-1}\widehat{M}_{21}\delta_{p_0} , \qquad B_-^r = 2\pi\varpi(p_0)\widehat{M}_{22}^{-1}\delta_{p_0} . \tag{68}$$

$$A_+^l = 2\pi\varpi(p_0)\big(\widehat{M}_{11} - \widehat{M}_{12}\widehat{M}_{22}^{-1}\widehat{M}_{21}\big)\delta_{p_0} , \qquad A_+^r = 2\pi\varpi(p_0)\widehat{M}_{12}\widehat{M}_{22}^{-1}\delta_{p_0} , \tag{69}$$

substituting (65) and (66) in these equations, and using the result in (13), (14), (15) and (16), we obtain a formal solution of the scattering problem for the potential (1). In the following, we pursue an alternative route for solving this problem which aims at computing the scattering matrix $\widehat{\mathbf{S}}$.

First, we introduce

$$\check{A}_- := \widehat{\varpi}^{-1}e^{ia_-\widehat{\varpi}}A_- , \qquad \check{\mathscr{B}}_- := \widehat{\varpi}^{-1}e^{-ia_-\widehat{\varpi}}\mathscr{B}_- , \tag{70}$$

$$\check{\mathscr{A}}_+ := \widehat{\varpi}^{-1}e^{ia_+\widehat{\varpi}}\mathscr{A}_+ , \qquad \check{B}_+ := \widehat{\varpi}^{-1}e^{-ia_+\widehat{\varpi}}B_+ , \tag{71}$$

$$\widehat{\mathbf{Q}} := (\widehat{W} - \widehat{\varpi})\mathbf{I} + (\widehat{W} + \widehat{\varpi})\boldsymbol{\sigma}_1 = \begin{bmatrix} \widehat{W} - \widehat{\varpi} & \widehat{W} + \widehat{\varpi} \\ \widehat{W} + \widehat{\varpi} & \widehat{W} - \widehat{\varpi} \end{bmatrix} , \tag{72}$$

and use (49) to establish the intertwining relation,

$$\widehat{\mathbf{Q}} \, \widehat{\varpi}^{-1}e^{-ix\widehat{\mathbf{H}}}\widehat{\varpi} = e^{-ix\widehat{W}\boldsymbol{\sigma}_3}\widehat{\mathbf{Q}} . \tag{73}$$

According to (70) and (71),

$$\begin{bmatrix} A_- \\ \mathscr{B}_- \end{bmatrix} = e^{-ia_-\widehat{\varpi}\boldsymbol{\sigma}_3}\widehat{\varpi} \begin{bmatrix} \check{A}_- \\ \check{\mathscr{B}}_- \end{bmatrix} , \qquad \begin{bmatrix} \mathscr{A}_+ \\ B_+ \end{bmatrix} = e^{-ia_+\widehat{\varpi}\boldsymbol{\sigma}_3}\widehat{\varpi} \begin{bmatrix} \check{\mathscr{A}}_+ \\ \check{B}_+ \end{bmatrix} ,$$

If we substitute these relations together with (34) in (22), we find

$$\widehat{\varpi}^{-1}e^{-ia\widehat{\mathbf{H}}}\widehat{\varpi} \begin{bmatrix} \check{A}_- \\ \check{\mathscr{B}}_- \end{bmatrix} = \begin{bmatrix} \check{\mathscr{A}}_+ \\ \check{B}_+ \end{bmatrix} .$$

In light of (73), applying $\widehat{\mathbf{Q}}$ to both sides of this equation yields

$$\widehat{\mathbf{Q}} \begin{bmatrix} \check{A}_- \\ \check{\mathscr{B}}_- \end{bmatrix} = e^{ia\widehat{W}\boldsymbol{\sigma}_3}\widehat{\mathbf{Q}} \begin{bmatrix} \check{\mathscr{A}}_+ \\ \check{B}_+ \end{bmatrix} .$$

With the help of (72), we can express this equation in the form

$$e^{\pm\frac{ia}{2}\widehat{W}}\big[(\widehat{W}\mp\widehat{\varpi})\check{\mathscr{A}}_+ + (\widehat{W}\pm\widehat{\varpi})\check{B}_+\big] = e^{\mp\frac{ia}{2}\widehat{W}}\big[(\widehat{W}\mp\widehat{\varpi})\check{A}_- + (\widehat{W}\pm\widehat{\varpi})\check{\mathscr{B}}_-\big].$$

It is not difficult to check that this is equivalent to

$$\widehat{\mathbf{Q}}_+\left[\begin{array}{c}\check{\mathscr{A}}_+\\[2pt]\check{\mathscr{B}}_-\end{array}\right] = \widehat{\mathbf{Q}}_-\left[\begin{array}{c}\check{A}_-\\[2pt]\check{B}_+\end{array}\right], \tag{74}$$

where

$$\widehat{\mathbf{Q}}_\pm := \left[\begin{array}{cc}e^{\pm ia\widehat{W}/2}(\widehat{W}-\widehat{\varpi}) & -e^{\mp ia\widehat{W}/2}(\widehat{W}+\widehat{\varpi})\\[4pt]e^{\mp ia\widehat{W}/2}(\widehat{W}+\widehat{\varpi}) & -e^{\pm ia\widehat{W}/2}(\widehat{W}-\widehat{\varpi})\end{array}\right]. \tag{75}$$

Next, we multiply both sides of (74) by the $1\times 2$ matrices $[1\ \ \pm 1]$ from the left and use (75) to show that

$$\widehat{\Omega}_{1-}(\check{\mathscr{A}}_+ - \check{\mathscr{B}}_-) = \widehat{\Omega}_{1+}(\check{A}_- - \check{B}_+), \qquad \widehat{\Omega}_{2-}(\check{\mathscr{A}}_+ + \check{\mathscr{B}}_-) = \widehat{\Omega}_{2+}(\check{A}_- + \check{B}_+), \tag{76}$$

where

$$\widehat{\Omega}_{1\pm} := \widehat{W}\cos(\tfrac{a}{2}\widehat{W}) \pm i\sin(\tfrac{a}{2}\widehat{W})\widehat{\varpi}, \qquad \widehat{\Omega}_{2\pm} := \cos(\tfrac{a}{2}\widehat{W})\widehat{\varpi} \pm i\widehat{W}\sin(\tfrac{a}{2}\widehat{W}). \tag{77}$$

Solving (76) for $\check{\mathscr{A}}_+$ and $\check{\mathscr{B}}_-$ and using (70) and (71) to express $\mathscr{A}_+$ and $\mathscr{B}_-$ in terms of $A_-$ and $B_+$, we obtain

$$\left[\begin{array}{c}\mathscr{A}_+\\[2pt]\mathscr{B}_-\end{array}\right] = \widehat{\varpi}\,\widehat{\mathbf{E}}_+\widehat{\boldsymbol{\Gamma}}\widehat{\mathbf{E}}_-\widehat{\varpi}^{-1}\left[\begin{array}{c}A_-\\[2pt]B_+\end{array}\right], \tag{78}$$

where

$$\widehat{\mathbf{E}}_\pm := \left[\begin{array}{cc}e^{\mp ia_\pm\widehat{\varpi}} & 0\\[4pt]0 & e^{\pm ia_\mp\widehat{\varpi}}\end{array}\right], \qquad \widehat{\boldsymbol{\Gamma}} := \left[\begin{array}{cc}\widehat{\Gamma}_+ & \widehat{\Gamma}_-\\[4pt]\widehat{\Gamma}_- & \widehat{\Gamma}_+\end{array}\right], \qquad \widehat{\Gamma}_\pm := \tfrac{1}{2}\big(\widehat{\Omega}_{1-}^{-1}\widehat{\Omega}_{1+} \pm \widehat{\Omega}_{2-}^{-1}\widehat{\Omega}_{2+}\big). \tag{79}$$

Comparing (78) with the second equation in (11) and making use of (9) we arrive at the following expression for the scattering matrix.

$$\widehat{\mathbf{S}} = \widehat{\boldsymbol{\Pi}}_k\,\widehat{\varpi}\,\widehat{\mathbf{E}}_+\widehat{\boldsymbol{\Gamma}}\widehat{\mathbf{E}}_-\widehat{\varpi}^{-1}. \tag{80}$$

Having determined the S-matrix, we can use (13), (13), and (20) to calculate the reflection, transmission, and scattering amplitudes of the potential. This requires the knowledge of the entries of $\langle p|\widehat{\mathbf{S}}|p_0\rangle$. Using (79) and (80) we can express these in terms of $\langle p|\widehat{\Gamma}_\pm|p_0\rangle$. Substituting the result in (21), setting

$$p_0 = k\sin\theta_0, \qquad\qquad p = k\sin\theta, \tag{81}$$

and making use of (20), we find

$$R^l(\theta) = i\sqrt{2\pi}\,k\cos\theta\,e^{ia_-k(\cos\theta_0-\cos\theta)}\,\Gamma_-(k\sin\theta, k\sin\theta_0), \tag{82}$$

$$T^l(\theta) = -i\sqrt{2\pi}\,k\cos\theta\,e^{ik(a_-\cos\theta_0-a_+\cos\theta)}\,\Gamma_+(k\sin\theta, k\sin\theta_0), \tag{83}$$

$$R^r(\theta) = -i\sqrt{2\pi}\,k\cos\theta\,e^{ia_+k(\cos\theta_0-\cos\theta)}\,\Gamma_-(k\sin\theta, k\sin\theta_0), \tag{84}$$

$$T^r(\theta) = i\sqrt{2\pi}\,k\cos\theta\,e^{ik(a_+\cos\theta_0-a_-\cos\theta)}\,\Gamma_+(k\sin\theta, k\sin\theta_0), \tag{85}$$

where

$$\Gamma_\pm(p, p_0) = \langle p|\widehat{\Gamma}_\pm|p_0\rangle, \tag{86}$$

and we have employed $\varpi(p) = k|\cos\theta|$ and $\varpi(p_0) = k|\cos\theta_0|$, and taken into account the relevant range of values of $\theta_0$ and $\theta$ in the expressions for $R^{l/r}(\theta)$ and $T^{l/r}(\theta)$.[5] Equations (13), (14), and (82) – (86) reduce the solution of the scattering problem for the potentials of the form (1) to the calculation of $\langle p|\widehat{\Gamma}_\pm|p_0\rangle$.

---

[5]For $R^l$, $\cos\theta_0 > 0 > \cos\theta$; for $T^l$, $\cos\theta_0 > 0 < \cos\theta$; for $R^r$, $\cos\theta_0 < 0 < \cos\theta$; for $T^r$, $R^l$, $\cos\theta_0 < 0 > \cos\theta$.

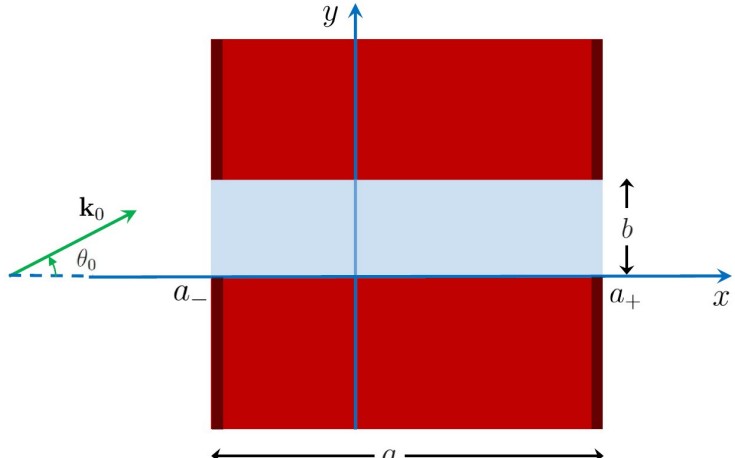

Figure 2: Schematic view of the scattering of a left-incident wave by a two-dimensional waveguide with length $a$, interior width $b$, and infinitely thick walls. The potential $v(x, y)$ becomes infinitely large inside the walls of the waveguide (regions colored in red), takes a constant value $\mathscr{V}_0$ in its interior (region colored in light blue), and vanishes elsewhere. The thick dark red lines signify the vertical boundaries of the guide which also contribute to the scattering of incident waves.

## 5  Application to a waveguide with a finite length

Consider the cases where $\mathscr{V}$ has the form,

$$\mathscr{V}(y) := \begin{cases} \mathscr{V}_0 & \text{for} \quad y \in [0, b], \\ +\infty & \text{for} \quad y \notin [0, b], \end{cases} \tag{87}$$

where $\mathscr{V}_0$ and $b$ are real parameters, and $b > 0$. Then the potential (1) describes the scattering of waves by a two-dimensional rectangular waveguide of length $a$ and width $b$ that contains a homogeneous inactive and lossless material and has impenetrable walls of infinite thickness. Fig. 2 shows a schematic view of such a waveguide. $\mathscr{V}_0$ determines the scattering properties of its content.

For this choice of $\mathscr{V}$, $\hat{p}^2 + \widehat{\mathscr{V}}$ is the Hamiltonian operator for a particle trapped in an infinite rectangular potential well. Usually, it is taken to act in the Hilbert space $L^2[0, b]$ of square-integrable functions defined on the interval $[0, b]$. Because this potential is an idealization of a finite potential well with high walls, it is more convenient to retain $L^2(\mathbb{R})$ as the Hilbert space of the system. To do this, first we recall the orthogonal direct sum decomposition of $L^2(\mathbb{R})$ given by [43],

$$L^2(\mathbb{R}) = L^2[0, b] \oplus L^2(\mathbb{R} \setminus [0, b]), \tag{88}$$

where for each $\mathcal{S} \subseteq \mathbb{R}$, $L^2(\mathcal{S})$ is the Hilbert space of sqaure-integrable functions $\xi : \mathcal{S} \to \mathbb{C}$. Equation (88) means that for each $\phi \in L^2(\mathbb{R})$ there are unique $\mathring{\phi} \in L^2[0, b]$ and $\check{\phi} \in L^2(\mathbb{R} \setminus [0, b])$ such that

$$\phi(y) = \begin{cases} \mathring{\phi}(y) & \text{for} \quad y \in [0, b], \\ \check{\phi}(y) & \text{for} \quad y \notin [0, b]. \end{cases} \tag{89}$$

Next, we let $\widehat{\Lambda}$ be the projection operator defined on $L^2(\mathbb{R})$ by

$$\left(\widehat{\Lambda}\phi\right)(x) := \begin{cases} \mathring{\phi}(y) & \text{for} \quad y \in [0, b], \\ 0 & \text{for} \quad y \notin [0, b], \end{cases} \tag{90}$$

and identify $\hat{p}^2 + \widehat{\mathscr{V}}$ for the infinite barrier potential (87) with the operator $\widehat{\Lambda}(\hat{p}^2 + \mathscr{V}_0\widehat{I})\widehat{\Lambda}$ that is defined by

$$\langle y|\left(\hat{p}^2 + \widehat{\mathscr{V}}\right)|\phi\rangle := \begin{cases} -\mathring{\phi}''(y) + \mathscr{V}_0\mathring{\phi}(y) & \text{for} \quad x \in [0, b], \\ 0 & \text{for} \quad x \notin [0, b], \end{cases} \tag{91}$$

on the domain,

$$\mathcal{D} := \left\{\phi \in L^2(\mathbb{R}) \mid \mathring{\phi}'' \in L^2[0, b], \; \mathring{\phi}(0) = \mathring{\phi}(b) = 0\right\}, $$

where $\mathring{\phi}''$ denotes the second derivative of $\mathring{\phi}$.

The above extension of the standard Hamiltonian operator for the infinite potential well (87) to the Hilbert space $L^2(\mathbb{R})$ adds an infinitely degenerate zero eigenvalue to its spectrum.[6] The determination of nonzero eigenvalues and a corresponding orthonormal set of eigenfunctions of this operator is an elementary exercise. They are respectively given by

$$E_n = \left(\frac{\pi n}{b}\right)^2 + \mathscr{V}_0, \qquad \phi_n(y) = \begin{cases} \sqrt{2/b} \; \sin(\pi n y/b) & \text{for} \quad y \in [0, b], \\ 0 & \text{for} \quad y \notin [0, b], \end{cases} \tag{92}$$

where $n \in \mathbb{Z}^+$. Notice however that the standard completeness relation for $\phi_n$ is replaced by

$$\sum_{n=1}^{\infty} |\phi_n\rangle\langle\phi_n| = \widehat{\Lambda}. \tag{93}$$

This is consistent with the fact that $\mathring{\phi}_n$ form an orthonormal basis of $L^2[0, b]$.

The operator $\widehat{\Lambda}$ is an orthogonal projection operator [42] whose range and null space (kernel) are respectively isomorphic to $L^2[0, b]$ and $L^2(\mathbb{R} \setminus [0, b])$. It is also clear that the range of $\widehat{\Lambda}$ coincides with the null space of $\widehat{I} - \widehat{\Lambda}$. Therefore, the latter is isomorphic to $L^2[0, b]$. Similarly, the range of $\widehat{I} - \widehat{\Lambda}$ which coincides with the null space of $\widehat{\Lambda}$ is isomorphic to $L^2(\mathbb{R} \setminus [0, b])$. For these reasons, in what follows we identify the range of $\widehat{\Lambda}$ and $\widehat{I} - \widehat{\Lambda}$ with $L^2[0, b]$ and $L^2(\mathbb{R} \setminus [0, b])$, respectively.

The scattering of waves by our waveguide system is due to two different interactions. A part of the wave enters the waveguide, interacts with the material inside it, and gets partly reflected and partly transmitted. The other part interacts with the impenetrable vertical boundaries of the guide (represented by the thick dark red lines in Fig. 2) and is reflected. Since the scattering phenomenon is defined by the Schrödinger equation which is linear, the scattered wave is the superposition of the contributions of the interior and vertical boundaries of the guide.

Consider a general bounded solution $\psi$ of the Schrödinger equation (2) which satisfies (5). For each $x \in \mathbb{R}$, we use $|\psi(x)\rangle$ to label the function $\psi(x, \cdot) : \mathbb{R} \to \mathbb{C}$, so that $\langle y|\psi(x)\rangle := \psi(x, y)$. This allows us to write (5) in the form,

$$|\psi(x)\rangle = \frac{1}{\sqrt{2\pi}} \times \begin{cases} \widehat{\varpi}^{-1}\left[e^{ix\widehat{\varpi}}|A_-\rangle + e^{-ix\widehat{\varpi}}|\mathscr{B}_-\rangle\right] & \text{for} \quad x \leq a_-, \\ \widehat{\varpi}^{-1}\left[e^{ix\widehat{\varpi}}|\mathscr{A}_+\rangle + e^{-ix\widehat{\varpi}}|B_+\rangle\right] & \text{for} \quad x \geq a_+, \end{cases} \tag{94}$$

where $|A_-\rangle, |\mathscr{B}_-\rangle, |\mathscr{A}_+\rangle$, and $|B_+\rangle$ respectively denote the coefficient functions $A_-, \mathscr{B}_-, \mathscr{A}_+$, and $B_+$, and we have made use of $\langle y|p\rangle = e^{ipy}/\sqrt{2\pi}$.

---

[6]This is because every smooth function $\phi$ that vanishes in $[0, b]$ belongs to $\mathcal{D}$ and satisfies $(\hat{p}^2 + \widehat{\mathscr{V}})|\phi\rangle = 0$.

Next, we use (92) and (93) to express the boundary conditions, $\psi(a_\pm, y) = 0$ for $y \notin [0, b]$, at the vertical boundaries of the waveguide as

$$(\widehat{I} - \widehat{\Lambda})|\psi(a_\pm)\rangle = 0. \tag{95}$$

Substituting (94) in this equation, we arrive at

$$(\widehat{\Lambda} - \widehat{I})|\mathscr{B}_-\rangle = -(\widehat{\Lambda} - \widehat{I})e^{2ia_-\widehat{\varpi}}|A_-\rangle, \tag{96}$$

$$(\widehat{\Lambda} - \widehat{I})|\mathscr{A}_+\rangle = -(\widehat{\Lambda} - \widehat{I})e^{-2ia_+\widehat{\varpi}}|B_+\rangle. \tag{97}$$

These are non-homogeneous linear equations for $|\mathscr{B}_-\rangle$ and $|\mathscr{A}_+\rangle$. We can express their general solution in the form

$$|\mathscr{B}_-\rangle = |\mathscr{B}_{0-}\rangle + (\widehat{I} - \widehat{\Lambda})e^{2ia_-\widehat{\varpi}}|A_-\rangle, \tag{98}$$

$$|\mathscr{A}_+\rangle = |\mathscr{A}_{0+}\rangle + (\widehat{I} - \widehat{\Lambda})e^{-2ia_+\widehat{\varpi}}|B_+\rangle, \tag{99}$$

where $|\mathscr{B}_{0-}\rangle$ and $|\mathscr{A}_{0+}\rangle$ represent the general solution of the homogeneous equation,

$$(\widehat{\Lambda} - \widehat{I})|\phi\rangle = 0 .$$

This means that $|\mathscr{B}_{0-}\rangle$ and $|\mathscr{A}_{0+}\rangle$ are associated with the Hilbert space $L^2[0, b]$. It is also clear that $(\widehat{I} - \widehat{\Lambda})e^{2ia_-\widehat{\varpi}}|A_-\rangle$ and $(\widehat{I} - \widehat{\Lambda})e^{-2ia_+\widehat{\varpi}}|B_+\rangle$ are associated with $L^2(\mathbb{R} \setminus [0, b])$.

For a left-incident wave, $A_-(p) = A_-^l(p) = 2\pi\varpi(p)\delta(p - p_0)$ and $B_+(p) = B_+^l(p) = 0$. Therefore, $|A_-\rangle = 2\pi\widehat{\varpi}|p_0\rangle$ and $|B_+\rangle = 0$. Substituting these in (98) and (99), applying $\widehat{\Pi}_k$, and employing (9), we obtain

$$|B_-^l\rangle = \widehat{\Pi}_k|\mathscr{B}_{0-}^l\rangle + 2\pi\varpi(p_0)e^{2ia_-\varpi(p_0)}\widehat{\Pi}_k(\widehat{I} - \widehat{\Lambda})|p_0\rangle, \tag{100}$$

$$|A_+^l\rangle = \widehat{\Pi}_k|\mathscr{A}_{0+}^l\rangle, \tag{101}$$

where we use the superscript "$l$" to emphasize that we consider left-incident waves. For $|p| < k$, (100) and (101) imply

$$B_-^l(p) = \mathscr{B}_{0-}^l(p) + B_{1-}^l(p), \qquad\qquad A_+^l(p) = \mathscr{A}_{0+}^l(p), \tag{102}$$

where

$$B_{1-}^l(p) := \varpi(p_0)e^{2ia_-\varpi(p_0)}\left[2\pi\delta(p - p_0) - \sum_{n=1}^{\infty}\tilde{\phi}_n(p_0)^*\tilde{\phi}_n(p)\right], \tag{103}$$

and $\tilde{\phi}_n(p)$ is the Fourier transform of $\phi_n(y)$, i.e.,

$$\tilde{\phi}_n(p) := \int_{-\infty}^{\infty} dy\, e^{-ipy}\phi_n(y) = \begin{cases} \dfrac{\pi n\sqrt{2b}[e^{-i(bp-\pi n)} - 1]}{(bp)^2 - (\pi n)^2} & \text{for } p \neq \pi n/b, \\[3mm] -i\sqrt{b/2} & \text{for } p = \pi n/b. \end{cases} \tag{104}$$

Note that $\tilde{\phi}_n(p) = \sqrt{2\pi}\langle p|\phi_n\rangle$.

Recalling that $B_-^l$ and $A_+^l$ respectively determine the left reflection and transmission amplitudes of the potential and that $\mathscr{B}_{0-}^l$ and $\mathscr{A}_{0+}^l$ are associated with the same Hilbert space as the one we use to describe the waves propagating inside the waveguide, we identify these functions with those that encode the contribution of the content of the waveguide to the reflection and transmission of the left-incident waves. Following the same reasoning $B_{1-}^l$ represents the contribution of the vertical boundary of the waveguide located on the line $x = a_-$ to the reflection of these waves. Clearly, the presence of the vertical boundary at $x = a_+$ does not affect the reflection or transmission of the left-incident waves.

The treatment of the right-incident waves is analogous. Repeating the analysis of the preceding section, we find

$$B_-^r(p) = \mathscr{B}_{0-}^r(p), \qquad\qquad A_+^r(p) = \mathscr{A}_{0+}^r(p) + A_{1+}^r(p), \qquad (105)$$

where $\mathscr{B}_{0-}^r$ and $\mathscr{A}_{0+}^r$ are coefficient functions belonging to $L^2[0,b]$ that represent the contribution of the content of the guide to the scattering of right-incident waves, and

$$A_{1+}^r(p) := \varpi(p_0) e^{-2ia_+ \varpi(p_0)} \left[ 2\pi\delta(p-p_0) - \sum_{n=1}^{\infty} \tilde{\phi}_n(p_0)^* \tilde{\phi}_n(p) \right]. \qquad (106)$$

Equations (13) – (16), (102), (103), (105), and (106) reduce the solution of the scattering problem for our waveguide to the determination of $\mathscr{B}_{0-}^{l/r}(p)$ and $\mathscr{A}_{0+}^{l/r}(p)$. We can compute these using the machinery developed in Sec. 4. More specifically, they are given by the right-hand side of (78). This in turn identifies the contribution of the content of the waveguide to the reflection and transmission amplitudes with the right-hand sides of (82) – (85). Adding the contribution of the vertical boundaries of the guide to the reflection amplitudes, which are stored in $B_{1-}^l$ and $A_{1+}^r$, we have

$$R^l(\theta) = -\frac{i}{\sqrt{2\pi}} B_{1-}^l(k\sin\theta) + i\sqrt{2\pi}\, k\cos\theta\, e^{ia_- k(\cos\theta_0 - \cos\theta)}\, \Gamma_-(k\sin\theta, k\sin\theta_0), \qquad (107)$$

$$R^r(\theta) = -\frac{i}{\sqrt{2\pi}} A_{1+}^r(k\sin\theta) - i\sqrt{2\pi}\, k\cos\theta\, e^{ia_+ k(\cos\theta_0 - \cos\theta)}\, \Gamma_-(k\sin\theta, k\sin\theta_0), \qquad (108)$$

where we have used (15), (82), (84), (102), and (105). Note also that the $B_{1-}^l(k\sin\theta)$ and $A_{1+}^r(k\sin\theta)$ appearing in (107) and (108), are respectively given by (103) and (106) with $p_0 = k\sin\theta_0$; they have the forms.

$$B_{1-}^l(k\sin\theta) = e^{2ia_- k\cos\theta_0}\Big[ 2\pi\delta(\theta-\theta_0) - k\cos\theta_0 \sum_{n=1}^{\infty} \tilde{\phi}_n(k\sin\theta_0)^* \tilde{\phi}_n(k\sin\theta) \Big], \qquad (109)$$

$$A_{1+}^r(k\sin\theta) = e^{2ia_+ k\cos\theta_0}\Big[ 2\pi\delta(\theta-\theta_0) + k\cos\theta_0 \sum_{n=1}^{\infty} \tilde{\phi}_n(k\sin\theta_0)^* \tilde{\phi}_n(k\sin\theta) \Big]. \qquad (110)$$

Because the transmission amplitudes $T^{l/r}(\theta)$ do not get affected by the presence of the vertical boundaries of the waveguide, they are still given by (83) and (85).

Next, we explore the contribution of the interior of the waveguide. The waves propagating inside the waveguide are described by functions vanishing outside $[0,b]$. This suggests that the operator $\widehat{W}$ associated with our waveguide system is given by (42) with $\phi_n$'s having the form (92). In view of (93), this implies

$$\widehat{W}\widehat{\Lambda} = \widehat{\Lambda}\widehat{W} = \widehat{W}. \qquad (111)$$

In particular, $L^2(\mathbb{R}\setminus[0,b])$ is a subset of the kernel of $\widehat{W}$. This identifies zero as an infinitely degenerate eigenvalue of $\widehat{W}$. We can use (39) and the first equation in (92) to determine other eigenvalues of $\widehat{W}$. They are given by

$$w_n = \begin{cases} \sqrt{k^2 - (\pi n/b)^2 - \mathscr{V}_0} & \text{for} \quad k^2 > \mathscr{V}_0 \ \text{and} \ n \le b\sqrt{k^2 - \mathscr{V}_0}/\pi, \\ i\sqrt{(\pi n/b)^2 + \mathscr{V}_0 - k^2} & \text{otherwise.} \end{cases} \qquad (112)$$

Fig. 3 shows plots of the real and imaginary parts of the eigenvalues $\pm w_n$ of the effective Hamiltonian $\widehat{\mathbf{H}}$ as functions of $k$ for $n = 1, 2, 3, 4$, $\mathscr{V}_0 = 0$ (empty waveguide) and $\mathscr{V}_0 = -(5\pi/2b)^2$.

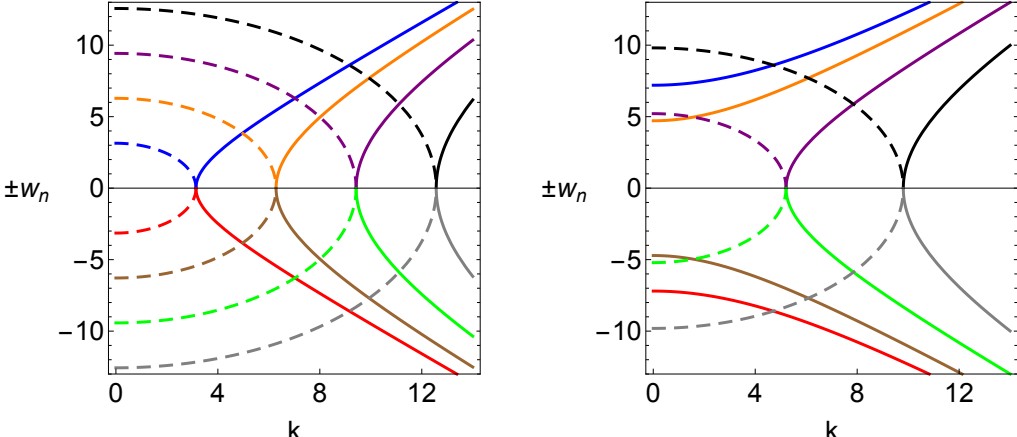

Figure 3: Plots of the real and imaginary parts of the eigenvalues $\pm w_n$ of $\widehat{\mathbf{H}}$ as functions of $k$ for $n = 1, 2, 3, 4$, $\mathcal{V}_0 = 0$ (on the left) and $\mathcal{V}_0 = -(5\pi/2b)^2$ (on the right). The solid and dashed curves respectively correspond to the real and imaginary parts of $\pm w_n$. Values of $\pm w_n$ and $k$ are in units of $b^{-1}$. The blue (resp. red), orange (resp. brown), purple (resp. green), and black (resp. gray) curves correspond to $w_n$ (resp. $-w_n$) with $n = 1, 2, 3$, and $4$, respectively. Their crossing points with the $k$-axis mark the exceptional points. For $\mathcal{V}_0 = -(5\pi/2b)^2$, there are no exceptional points for $n = 1, 2$.

The points on the $k$-axis where the graphs of $\pm w_n$ intersect represent the exceptional points of $\mathbf{H}$. The corresponding (exceptional) wave numbers $k_n$ form an increasing sequence. For $\mathcal{V}_0 = 0$ and $k < k_n$, $\pm w_n$ are purely imaginary, and as we increase $k$, they approach and collide at an exceptional point and then separate as a pair of real eigenvalues having opposite sign. This holds also for $\mathcal{V}_0 \neq 0$, except that when $\mathcal{V}_0 < -(\pi/b)^2$ and $n \leq b\sqrt{|\mathcal{V}_0|}/\pi$, $\pm w_n$ take real values for all $k$. Therefore, no exceptional points arises for these values of $n$. This is also depicted in Fig. 3; when $\mathcal{V}_0 = -(5\pi/2b)^2$, exceptional points are absent for $n = 1$ and $n = 2$.

To identify the proper definition of the operator $\widehat{\varpi}$ that enters the calculation of the reflection and transmission amplitudes due to the interior of the waveguide, we reexamine the role of the two-component wave function $\mathbf{\Psi}(x)$ in Sec. 3. Using $|\mathbf{\Psi}(x)\rangle$ to denote this wave function, we can express (29) as

$$|\mathbf{\Psi}(x)\rangle := \frac{1}{2} e^{-ix\widehat{\varpi}\sigma_3} \left[ \begin{array}{c} \widehat{\varpi}|\psi(x)\rangle - i\partial_x|\psi(x)\rangle \\ \widehat{\varpi}|\psi(x)\rangle + i\partial_x|\psi(x)\rangle \end{array} \right]. \tag{113}$$

Let us confine our attention to values of $x$ that lie in the interval $(a_-, a_+)$. Then $\psi(x, y) = 0$ for $y \notin [0, b]$. This means that $\widehat{\Lambda}|\psi(x)\rangle = |\psi(x)\rangle$. Therefore, $|\psi(x)\rangle$ belongs to the range of $\widehat{\Lambda}$ which we identify with $L^2[0, b]$. In light of (113), the same applies to the components of $|\mathbf{\Psi}(x)\rangle$. This observation together with the requirement that, for $x \in (a_-, a_+)$, the time-independent Schrödinger equation (2) be equivalent to the time-dependent Schrödinger equation (28) suggests setting

$$\widehat{\varpi} := \widehat{\Lambda}\,\varpi(\hat{p})\widehat{\Lambda}, \tag{114}$$

and

$$\widehat{\mathcal{H}}(x) := \frac{\mathcal{V}_0}{2}\widehat{\Lambda}\,e^{-i\widehat{\varpi}x\sigma_3}\mathcal{K}\,e^{i\widehat{\varpi}x\sigma_3}\varpi(\hat{p})^{-1}\widehat{\Lambda} \quad \text{for} \quad x \in (a_-, a_+), \tag{115}$$

where we have also made use of (1) and (87). Enforcing (114) and (115), we can apply the constructions of Secs. 3 and 4 to describe the propagation of waves inside the waveguide.

According to (114),

$$\widehat{\Lambda}\widehat{\varpi}(\widehat{I}-\widehat{\Lambda}) = (\widehat{I}-\widehat{\Lambda})\widehat{\varpi}\widehat{\Lambda} = \widehat{0}, \tag{116}$$

which in particular implies

$$[\widehat{\varpi},\widehat{\Lambda}] = \widehat{0}. \tag{117}$$

Furthermore, in light of (6), (26), (92), (114), and the fact that for $y \in [0, b]$,

$$\langle y|\hat{p}^2|\mathring{\phi}_n\rangle = -\mathring{\phi}_n''(y) = -\phi_n''(y) = (\pi n/b)^2 \phi_n(y) = (\pi n/b)^2 \mathring{\phi}_n(y),$$

we have

$$\langle y|\widehat{\varpi}|\phi_n\rangle = \begin{cases} \varpi_n\mathring{\phi}_n(y) & \text{for} \quad y \in [0, b], \\ 0 & \text{for} \quad y \notin [0, b], \end{cases} \tag{118}$$

where

$$\varpi_n := \begin{cases} \sqrt{k^2-(\pi n/b)^2} & \text{for} \quad n \leq bk/\pi, \\ i\sqrt{(\pi n/b)^2-k^2} & \text{for} \quad n > bk/\pi. \end{cases} \tag{119}$$

Equations (92) and (118) show that

$$\widehat{\varpi}|\phi_n\rangle = \varpi_n|\phi_n\rangle. \tag{120}$$

An important consequence of (42), (93), (111), (116), and (120) is

$$\begin{aligned}
[\widehat{W},\widehat{\varpi}] &= \widehat{W}\widehat{\varpi} - \widehat{\varpi}\widehat{W} \\
&= \widehat{W}\widehat{\Lambda}\widehat{\varpi}(\widehat{I}-\widehat{\Lambda}) + \widehat{W}\widehat{\Lambda}\widehat{\varpi}\widehat{\Lambda} - \widehat{\varpi}\widehat{W} \\
&= \widehat{W}\widehat{\varpi}\widehat{\Lambda} - \widehat{\varpi}\widehat{W} \\
&= \widehat{W}\widehat{\varpi}\sum_{n=1}^{\infty}|\phi_n\rangle\langle\phi_n| - \widehat{\varpi}\sum_{n=1}^{\infty}w_n|\phi_n\rangle\langle\phi_n| \\
&= \sum_{n=1}^{\infty}(\varpi_n\widehat{W} - w_n\widehat{\varpi})|\phi_n\rangle\langle\phi_n| \\
&= \widehat{0}.
\end{aligned} \tag{121}$$

This relation simplifies the calculation of $\widehat{\Gamma}_{\pm}$ considerably. Using (77), (79), (111), and (121), we find

$$\widehat{\Gamma}_+ = \left[\frac{4\widehat{\varpi}\widehat{W}e^{ia\widehat{W}}}{(\widehat{W}+\widehat{\varpi})^2 - (\widehat{W}-\widehat{\varpi})^2 e^{2ia\widehat{W}}}\right]\widehat{\Lambda}, \tag{122}$$

$$\widehat{\Gamma}_- = \left[\frac{(\widehat{W}^2-\widehat{\varpi}^2)(\widehat{I}-e^{2ia\widehat{W}})}{(\widehat{W}+\widehat{\varpi})^2 - (\widehat{W}-\widehat{\varpi})^2 e^{2ia\widehat{W}}}\right]\widehat{\Lambda}. \tag{123}$$

It is not difficult to see that these relations hold also for the exceptional wavenumbers, if $\mathcal{V}_0 \neq 0$. We can derive the corresponding relations for the cases where $\mathcal{V}_0 = 0$ and $k$ is an exceptional wavenumber from (122) and (123) by taking their $\mathcal{V}_0 \to 0$ limit. We will examine the role and consequences of setting $k$ to one of its exceptional values at the end of this section.

In order to elucidate the physical implications of (122) and (123), first we consider incident waves for which $k^2 \geq \pi^2/b^2 + \mathcal{V}_0$, i.e., $k^2$ is not smaller than the ground state energy of the infinite barrier potential (87). Let $n_\star$ denote the integer part of $b\sqrt{k^2-\mathcal{V}_0}/\pi$, i.e.,

$$n_\star := \left\lfloor \frac{b}{\pi}\sqrt{k^2-\mathcal{V}_0} \right\rfloor. \tag{124}$$

Then, $n_\star \geq 1$, and according to (112), $w_n = |w_n|$ for $n \leq n_\star$, and $w_n = i|w_n| \neq 0$ for $n > n_\star$. This together with (42) and (120) allow us to express $\widehat{\Gamma}_\pm$ in the form,

$$\widehat{\Gamma}_\pm = \sum_{n=1}^{n_\star} r_n^\pm |\phi_n\rangle\langle\phi_n| + \sum_{n=n_\star+1}^{\infty} s_n^\pm |\phi_n\rangle\langle\phi_n|, \tag{125}$$

where

$$r_n^+ := \frac{4|w_n|\varpi_n e^{ia|w_n|}}{(\varpi_n + |w_n|)^2 - (\varpi_n - |w_n|)^2 e^{2ia|w_n|}}, \quad s_n^+ := \frac{4i|w_n|\varpi_n e^{-a|w_n|}}{(\varpi_n + i|w_n|)^2 - (\varpi_n - i|w_n|)^2 e^{-2a|w_n|}}, \tag{126}$$

$$r_n^- := \frac{(|w_n|^2 - \varpi_n^2)(1 - e^{2ia|w_n|})}{(\varpi_n + |w_n|)^2 - (\varpi_n - |w_n|)^2 e^{2ia|w_n|}}, \quad s_n^- := -\frac{(|w_n|^2 + \varpi_n^2)(1 - e^{-2a|w_n|})}{(\varpi_n + i|w_n|)^2 - (\varpi_n - i|w_n|)^2 e^{-2a|w_n|}}. \tag{127}$$

Inserting (125) in (86), we have

$$\Gamma_\pm(p, p_0) = \frac{1}{2\pi}\left[\sum_{n=1}^{n_\star} r_n^\pm \tilde{\phi}_n(p_0)^* \tilde{\phi}_n(p) + \sum_{n=n_\star+1}^{\infty} s_n^\pm \tilde{\phi}_n(p_0)^* \tilde{\phi}_n(p)\right]. \tag{128}$$

In the appendix, we show that whenever $k^2 \geq (\pi/b)^2 + \mathcal{V}_0$ and $n > n_\star$,

$$a|w_n| > \frac{\sqrt{2}\,\pi a\,\eta(k)}{b}, \tag{129}$$

where

$$\eta(k) := \sqrt{n_\star + 1 - \frac{b}{\pi}\sqrt{k^2 - \mathcal{V}_0}}.$$

Notice that by virtue of (124), $0 < \eta(k) \leq 1$. According to (129), if the waveguide's length is so much larger than its width that $a\eta(k)/b \gg 1$, then $a|w_n| \gg 1$ for $n > n_\star$. In this case, we can use (126) – (128) to obtain the following approximate expressions for $\Gamma_\pm(p, p_0)$.

$$\Gamma_+(p, p_0) \approx \frac{1}{2\pi}\sum_{n=1}^{n_\star} r_n^+ \tilde{\phi}_n(p_0)^* \tilde{\phi}_n(p), \tag{130}$$

$$\Gamma_-(p, p_0) \approx \frac{1}{2\pi}\left[\sum_{n=1}^{n_\star} r_n^- \tilde{\phi}_n(p_0)^* \tilde{\phi}_n(p) + \sum_{n=n_\star+1}^{\infty} t_n \tilde{\phi}_n(p_0)^* \tilde{\phi}_n(p)\right], \tag{131}$$

where

$$t_n := \frac{|w_n| + i\varpi_n}{|w_n| - i\varpi_n}. \tag{132}$$

According to (83), (85), and (130), the transmission of high-energy waves by a finite-length waveguide is determined by the first $n_\star$ energy eigenvalues and eigenfunctions of the infinite potential well (87), if $k^2 \geq \pi^2/b^2 + \mathcal{V}_0$. In the limit $a \to \infty$, the approximate relations (130) and (131) become exact equalities and agree with the fact that an infinitely long rectangular waveguide has finitely many propagating modes.

Next, we consider situations where $k^2 < \pi^2/b^2 + \mathcal{V}_0$. Then $w_n = i|w_n|$ for all $n \in \mathbb{Z}^+$, and (122) and (123) take the form,

$$\widehat{\Gamma}_\pm = \sum_{n=1}^{\infty} s_n^\pm |\phi_n\rangle\langle\phi_n|. \tag{133}$$

We can then use (86), (112), (126), (127), and (132) to infer that whenever

$$\mathcal{V}_0 > \frac{1}{a^2} - \frac{\pi^2}{b^2} \quad \text{and} \quad k \ll \sqrt{\mathcal{V}_0 + \frac{\pi^2}{b^2} - \frac{1}{a^2}}, \tag{134}$$

we have

$$\Gamma_+(p, p_0) \approx 0, \qquad \Gamma_-(p, p_0) \approx \frac{1}{2\pi} \sum_{n=1}^{\infty} t_n \, \tilde{\phi}_n(p_0)^* \tilde{\phi}_n(p). \tag{135}$$

In light of (83) and (85), the first of these relations shows that the waveguide does not transmit the waves, i.e., it acts as a filter, if $\mathcal{V}_0$ and $k$ satisfy (134).

If the waveguide is empty, i.e., $\mathcal{V}_0 = 0$, $\widehat{W} = \widehat{\varpi}$, $w_n = \varpi_n$, and (122) and (123) become

$$\widehat{\Gamma}_+ = e^{ia\widehat{W}} \widehat{\Lambda}, \qquad \widehat{\Gamma}_- = \widehat{0}. \tag{136}$$

Substituting these in (86), we have

$$\Gamma_+(p, p_0) = \frac{1}{2\pi} \sum_{n=1}^{\infty} e^{ia\varpi_n} \tilde{\phi}_n(p_0)^* \tilde{\phi}_n(p), \qquad \Gamma_-(p, p_0) = 0, \tag{137}$$

where we have made use of (42) and (93). In view of (107) and (108), the second equation in (137) is consistent with the fact that the reflection of an incident wave from an empty waveguide is solely due to its vertical boundaries. Furthermore, when $1 \ll ak < \pi a/b$, $\varpi_n = i\sqrt{(\pi n/b)^2 - k^2}$ and all the terms contributing to $\Gamma_+(p, p_0)$ become exponentially small. This shows that the system acts as a filter for incident waves with such wavenumbers. If $k > \pi/b$ and $ak \gg 1$, then

$$\Gamma_+(p, p_0) \approx \frac{1}{2\pi} \sum_{n=1}^{\lfloor bk/\pi \rfloor} e^{iak\sqrt{1-(\pi n/bk)^2}} \tilde{\phi}_n(p_0)^* \tilde{\phi}_n(p). \tag{138}$$

Therefore, the transmission amplitudes are determined by the Fourier transform of $\phi_n$ for $n \le \lfloor bk/\pi \rfloor$.

Equations (136) turn out to hold also for exceptional wavenumbers $k_{n_\star}$. If $k = k_1 = \pi/b$, i.e., $k$ equals the smallest exceptional wavenumber, and $a \gg b$, then $ak \gg 1$, (137) holds, and (138) gives

$$\Gamma_+(p, p_0) \approx \frac{1}{2\pi} \tilde{\phi}_1(p_0)^* \tilde{\phi}_1(p). \tag{139}$$

According to (83) and (85), this shows that the intensity of the transmitted wave does not depend on the length of the waveguide $a$; it is invariant under continuous changes of $a$. This is a physical consequence of the presence of an exceptional point in our scattering setup. It is not difficult to see that the same phenomenon occurs if the waveguide is filled with a homogeneous material so that $\mathcal{V}_0$ takes a nonzero real value. In this case, for $a \gg b$ and $k = \sqrt{(\pi/b)^2 + \mathcal{V}_0}$, we have $w_1 = 0$, $a|w_n| \gg 1$ for $n \ge 2$, (126) gives $r_1^+ = 1$, and (130) reduces to (139).

In general, if $k$ is an exceptional wavenumber, so that $b\sqrt{k^2 - \mathcal{V}_0}/\pi$ is a positive integer, we have $n_\star = b\sqrt{k^2 - \mathcal{V}_0}/\pi$, $w_{n_\star} = 0$, and

$$\varpi_{n_\star} = \begin{cases} \sqrt{\mathcal{V}_0} & \text{for} \quad \mathcal{V}_0 \ge 0, \\ i\sqrt{|\mathcal{V}_0|} & \text{for} \quad \mathcal{V}_0 < 0. \end{cases} \tag{140}$$

We can describe the behavior of the reflection and transmission amplitudes in this case, by examining the limit when $k$ approaches the exceptional wavenumber $\sqrt{(\pi n_\star/b)^2 + \mathcal{V}_0}$. This

corresponding to evaluating the $w_{n_\star} \to 0$ limit of the formulas we have derived for the reflection and transmission amplitudes for non-exceptional wavenumbers, i.e., (83), (85), (107), and (108) with $\Gamma_\pm$ given by (128). Clearly, this only affects the contribution of the mode number $n_\star$ to $\Gamma_\pm$. Performing this limit in (126) and (127), we find

$$r_{n_\star}^+ = \frac{1}{1 - \frac{ia\varpi_{n_\star}}{2}}, \qquad\qquad r_{n_\star}^- = \frac{\frac{ia\varpi_{n_\star}}{2}}{1 - \frac{ia\varpi_{n_\star}}{2}} = r_{n_\star}^+ - 1. \tag{141}$$

Therefore, the presence of an exceptional point contributes the terms $r_{n_\star}^\pm \tilde\phi_{n_\star}(p_0)^* \tilde\phi_{n_\star}(p)/2\pi$ to $\Gamma_\pm(p, p_0)$. In view of (83), (85), (107), and (108), these correspond to the presence of terms in the reflection and transmission amplitudes that are rational functions of the length of the waveguide. For $\mathcal{V}_0 = 0$, $\varpi_{n_\star}(0) = 0$ and these terms become $a$-independent.

   Another consequence of the above analysis is that if we arrange to inject a wave to the waveguide from the left such that $A_-^l$ is proportional to $\phi_{n_\star}$ (and $B_+^l = 0$), then this wave will propagate through the guide in such a way that the transmitted wave only acquires a multiplicative factor given by $e^{-ia\varpi_{n_\star}}/[1 - ia\varpi_{n_\star}/2]$, i.e.,

$$\mathscr{A}_+^l = \frac{e^{-ia\varpi_{n_\star}}}{1 - \frac{ia\varpi_{n_\star}}{2}} A_-^l. \tag{142}$$

To see this, we use (78), (79), (120), and (125) to express the S-matrix associated with the interior of the waveguide in the form,

$$\mathbf{S}_0 = \sum_{n=1}^\infty \begin{bmatrix} e^{-ia\varpi_n}\Gamma_{+n} & e^{-2ia_+\varpi_n}\Gamma_{-n} \\ e^{2ia_-\varpi_n}\Gamma_{-n} & e^{-ia\varpi_n}\Gamma_{+n} \end{bmatrix} |\phi_n\rangle\langle\phi_n|, \tag{143}$$

where

$$\Gamma_{\pm n} := \langle\phi_n|\widehat{\Gamma}_\pm|\phi_n\rangle = \begin{cases} r_n^\pm & \text{for} \quad n \le n_\star, \\ s_n^\pm & \text{for} \quad n > n_\star. \end{cases} \tag{144}$$

When $A_-^l$ is proportional to $\phi_n$ and $B_+^l = 0$, $(\widehat{I} - \widehat\Lambda)|A_-^l\rangle = 0$, and (98) and (99) imply $\mathscr{B}_-^l = \mathscr{B}_{0-}^l$ and $\mathscr{A}_+^l = \mathscr{A}_{0+}^l$. We can use these relations together with (143) to conclude that

$$\begin{bmatrix} \mathscr{A}_+^l \\ \mathscr{B}_-^l \end{bmatrix} = \mathbf{S}\begin{bmatrix} A_-^l \\ 0 \end{bmatrix} = \mathbf{S}_0\begin{bmatrix} A_-^l \\ 0 \end{bmatrix} = \begin{bmatrix} e^{-ia\varpi_n}\Gamma_{+n} \\ e^{2ia_-\varpi_n}\Gamma_{-n} \end{bmatrix} A_-^l. \tag{145}$$

Equation (142) follows from (141), (144), and (145). If the waveguide is empty, $\varpi_{n_\star} = w_{n_\star} = 0$, and (142) becomes $\mathscr{A}_+^l = A_-^l$. This shows that the transmitted wave is identical to the injected wave both in amplitude and phase. There is also no reflected wave. Therefore, the waveguide does not scatter the injected wave.

## 6  Concluding remarks

Exceptional points are exclusive features of non-Hermitian operators. Therefore they do not appear in the standard formulation of quantum mechanics of closed systems where the observables and Hamiltonian are required to be Hermitian operators. Quantum mechanics may be formulated using certain non-Hermitian operators that are related to Hermitian operators via a similarity transformation [41], but these Hermitizable pseudo-Hermitian operators cannot support exceptional points either. These observations support the view that exceptional points do not play any role in quantum mechanics of closed systems. In the present paper, we have

offered a concrete evidence to the contrary. The stationary quantum scattering theory admits an interesting dynamical formulation allowing for a fundamental generalization of the notion of transfer matrix to dimensions larger than one. This is a linear operator acting in an infinite-dimensional function space that admits an expression in terms of the evolution operator for a non-Hermitian effective Hamiltonian operator. Even for cases where the scattering potential is real, this operator may possess exceptional points.

In this article, we have considered the application of the dynamical formulation of stationary scattering in two dimensions to a class of real potentials that do not vanish or decay to zero in an infinite region of the space, yet they define a valid scattering problem. For these potentials we have expressed the transfer matrix in terms of the evolution operator of a non-Hermitian effective Hamiltonian operator $\widehat{\mathbf{H}}$, determined the spectral properties of this operator, shown that it is a pseudo-Hermitian operator having a non-real spectrum, and calculated the transfer and scattering matrices. We have then confined our attention to the study of the scattering of plane waves by a finite-size waveguide with infinitely thick walls and investigated the contributions of the real and complex eigenvalues of $\widehat{\mathbf{H}}$ and its exceptional points to the scattering data. In particular, for an empty waveguide, we have shown that at the exceptional wavenumbers, where $\widehat{\mathbf{H}}$ develops an exceptional point, the transmitted wave includes a term that is independent of the length of the waveguide. This might find applications in calibration and sensing.

Our analysis may be applied to situations where the waveguide is filled with a homogeneous active or lossy material. This corresponds to situations where the constant $\mathcal{V}_0$ is complex. In this case, the eigenvalues of the infinite potential well (87) gets shifted by a complex constant but its eigenfunctions remain the same. Therefore, we can pursue the same approach to determine the transfer and scattering matrices for the problem. The main difference is that in this case, $w_n \neq 0$, and $\widehat{\mathbf{H}}$ no longer admits an exceptional point. This is indeed surprising, because exceptional points are available when the potential is real; they disappear when it becomes complex!

## Acknowledgements

This work has been supported by the Scientific and Technological Research Council of Turkey (TÜBİTAK) in the framework of the project 120F061 and by Turkish Academy of Sciences (TÜBA).

## Appendix: Derivation of (129)

Suppose that $k^2 \geq (\pi/b)^2 + \mathcal{V}_0$ and $n \geq n_\star + 1$. Then (112) and (124) imply

$$
\begin{aligned}
|w_n|^2 &= (\pi n/b)^2 + \mathcal{V}_0 - k^2 \\
&\geq (\pi/b)^2(n_\star + 1)^2 + \mathcal{V}_0 - k^2 \\
&\geq (\pi n_\star/b)^2 + \mathcal{V}_0 - k^2 + (\pi/b)^2(2n_\star + 1).
\end{aligned}
\tag{146}
$$

In view of (124), we also have

$$
n_\star > (b/\pi)\sqrt{k^2 - \mathcal{V}_0} - 1 \geq 0,
\tag{147}
$$

which implies

$$
(\pi n_\star/b)^2 > \left[\sqrt{k^2 - \mathcal{V}_0} - (\pi/b)\right]^2.
$$

Writing this relation in the form

$$(\pi n_\star/b)^2 + \mathcal{V}_0 - k^2 > -2(\pi/b)\sqrt{k^2 - \mathcal{V}_0} + (\pi/b)^2,$$

combining it with (146), and using (147), we find

$$|w_n|^2 \;\; > \;\; 2(\pi/b)^2 \left[ n_\star + 1 - (b/\pi)\sqrt{k^2 - \mathcal{V}_0} \right] > 0.$$

This implies (129).

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
