# Peer review of "Exceptional points and pseudo-Hermiticity in real potential scattering"

_SciPost Physics, doi:SciPost Phys. 12, 109 (2022)_

## Round 1 · Referee Report · Anonymous · 2021-11-23

Strengths
1. The manuscript is written in a clear, detailed, and self-contained manner.
2. The general results are well supported by the detailed investigation of a specific example.
3. The discovery that exceptional points are relevant even to the scattering problem of real potentials is interesting and may lead to further applications.
Weaknesses
1. The readability of the manuscript may be low for general readers (the manuscript is written in a self-contained manner, though).
Report
The authors develop a scattering theory in two dimensions for a class of potentials described by Eq. (1). They demonstrate that the transfer matrix for this scattering problem effectively reduces to a time-evolution operator for a non-Hermitian Hamiltonian with pseudo-Hermiticity even though the original scattering problem is concerned with a Hermitian Hamiltonian. As an example, they apply their general results to a specific waveguide of finite length. For this example, they show that the intensity of the transmitted wave is invariant under the continuous change of the length of the waveguide as a consequence of an exceptional point in the effective non-Hermitian Hamiltonian.
In my humble opinion, this manuscript discovers a new important role of exceptional points in the scattering theory, which opens a new pathway in non-Hermitian physics. Thus, I believe that this manuscript meets the acceptance criteria of SciPost Physics and I would like to recommend publication of this manuscript in SciPost Physics.
Requested changes
I have a couple of relatively minor comments, explained below.
1. As pointed out in the first observation in Sec. 3 (page 8), the spectrum of the effective non-Hermitian Hamiltonian consists of pairs of eigenvalues with opposite signs. This property should not originate from pseudo-Hermiticity since pseudo-Hermiticity only leads to real eigenvalues or complex-conjugate pairs of eigenvalues. Then, what symmetry ensures the opposite-sign pairs of eigenvalues? It seems to me that this structure originates from another internal symmetry such as particle-hole symmetry and sublattice symmetry [please see, for example, K. Kawabata et al., Phys. Rev. X 9, 041015 (2019) for details on internal symmetry of non-Hermitian Hamiltonians].
2. As described below Eq. (139) in Sec. 5 (page 20), a physical consequence of exceptional points in the scattering theory is the invariance of the intensity of the transmitted wave under the continuous change of the length of the waveguide. Is this property unique to exceptional points in the effective non-Hermitian Hamiltonian? In other words, can we realize the same phenomenon even if the effective Hamiltonian does not support exceptional points? Or, is it impossible to realize this phenomenon if the effective Hamiltonian does not support exceptional points?
Anonymous on 2021-11-27 [id 1981]
In the following we respond to the questions posed by the referee in her/his report.
1.~The referee asks about the origin of the structure of the spectrum of the effective non-Hermitian Hamiltonian~(33), namely the fact that its eigenvalues come in pairs of oppose sign. It is not difficult to show that the Pauli marix $\boldsymbol{\sigma}_1$ anti-commutes with the Hamiltonian, i.e., $\{\boldsymbol{\sigma}_1,\widehat{\mathbf{H}}\}={{\boldsymbol{0}}}$. Alternatively, according to (44), $\boldsymbol{\sigma}_1|\Psi_{n,\pm}{\rangle}=|\Psi_{n,\mp}{\rangle}$, which means that $\boldsymbol{\sigma}_1$ swaps the eigenvectors with eigenvalues differing by a sign. Therefore, if we use the terminology of K.~Kawabata et al, Phys. Rev. X {\bf 9}, 041015 (2019), we would say that this property of the spectrum of $\widehat{\mathbf{H}}$ follows from the chiral symmetry characterized by $\{\boldsymbol{\sigma}_1,\widehat{\mathbf{H}}\}={{\boldsymbol{0}}}$. This terminology does not however agree with the standard concept of symmetry (due to Wigner) which is described in terms of the commutation of ${\mathbf{H}}$ with either a linear or an antilinear operator. In view of Theorem~2 of Ref.~[38], the structure of the spectrum of $\widehat{\mathbf{H}}$ shows that both $\widehat{\mathbf{H}}$ and $i\widehat{\mathbf{H}}$ are pseudo-Hermitian and that each of them must commute with an antilinear involution. This implies the existence of antilinear involutions $\widehat{\mathfrak{S}}$ and $\widehat\chi$ satisfying
\begin{align}
&[\widehat{\mathbf{H}},\widehat{\mathfrak{S}}]=\widehat{{\boldsymbol{0}}},
&&\{\widehat{\mathbf{H}},\widehat\chi\}=\widehat{{\boldsymbol{0}}},
&&\widehat{\mathfrak{S}}^2=\widehat\chi^2=\widehat{\mathbf{I}}.
\nonumber
\end{align}
We can use the constructions given in Ref.~[38] to obtain spectral series expansions for $\widehat{\mathfrak{S}}$ and $\widehat\chi$. We do not report these in our paper, because we have not been able to find a useful physical application for them. We plan to add a remark on these observations in a revised version of our manuscript.
2.~The referee asks whether the invariance of the intensity of transmitted wave under continuous changes of the length of the waveguide is an exclusive consequence of the presence of an exceptional point. The answer to this question is in the affirmative. This can be seen from Eq.(137). For cases where there is no exceptional point, $w_{n}\neq 0$ for all $n$, and all the terms contributing to $\Gamma_+(p,p_0)$ become $a$-dependent.

---

## Round 1 · Referee Report · Anonymous · 2022-1-3

Strengths
1. The present work is thorough and rigorous.
2.This work complements the authors’ previous work on this topic.
3. The subject of exceptional points is interesting and timely.
4. Finding passive systems that can be used to demonstrate the existence of exceptional points is advantageous.
Weaknesses
1. The presentation is technical and contains many equations without explanations.
2. A discussion of the implication or significance of the results is lacking.
Report
I do not believe that this work meets the requirements of publication - that is making a groundbreaking discovery or opening a new field of research. I recommend condensing the text a bit and moving some of its derivations to the appendices, along with presenting some graphical illustrations of the results. My recommendation is that the paper should be revised significantly before acceptance in a more technical journal.
Requested changes
1. Are the exceptional points that you find related to the exceptional points associated with total internal reflection?
2. The authors use their recently developed scattering matrix formulation. I find its presentation difficult to follow. Since the formulation was published elsewhere, I recommend that the authors take this opportunity of writing a sequel paper on the same method to describe it in in an overview manner -- revealing the logic and essence of the approach. The authors can state the main equations one should use in order to construct the relevant operators in order to apply the method – rather than derive many results that were already published and present more than 100 equations.
3. I think it would be good to mention the advantages of the current formulation over previous scattering-matrix formulations of two-dimensional problems that require integration over momenta.
4. In Eq. 12, is p_0 defined anywhere? I think it is the incident momentum but could not find its definition.
5. The paper defines many quantities without explaining their physical meaning. To facilitate the reading, perhaps the authors can add explanations. For example, consider adding after the definition in Eq. (26) that it is an expansion of the identity operator in momentum space with weights w(p).
6. Is the number of real eigenvalues of H_hat (on page 8) equal to the number of bound states in the waveguide? Do the complex-conjugate pairs correspond to the continuum of unbounded solutions? What is the physical significance of their imaginary components? (penetration depths? lifetimes?)
7. I recommend plotting some of the results, perhaps using the studied numerical example. For example, the authors could plot the eigenvalues and eigenfunctions, showing that the spectrum contains exceptional points. Additionally, the authors could plot the dependence of scattering amplitudes on the geometrical or material parameters. Any plot of this sort and its discussion could have helped me appreciate the findings.
8. After Eq. 141, the manuscript makes an interesting point: “The presence of an exceptional points contributes terms … that correspond to the presence of terms in the reflection and transmission amplitudes that are rational functions of the length of the waveguide.“ Perhaps it could be mentioned in the abstract + introduction? If the authors plotted these reflection coefficients as a function of a parameter that drives the system into and out of the exceptional points, could they explain the behavior of the scattering amplitudes near the EPs I light of their analytic results in a graphical manner? Maybe plot these coefficients as a function of the length of the waveguide?
9. In the discussion, the authors say: “In particular, for an empty waveguide, we have shown that at the exceptional wavenumbers, where H develops an exceptional point, the transmitted wave includes a term that is independent of the length of the waveguide.“ Does this refer to the same point as my previous remark or is this a different result? Perhaps the authors could plot the contribution to the transmission that is independent and the one that depends on the length, and show how they relate to each other?
10. The authors say in the concluding paragraph that the addition of loss or gain would make the EPs disappear. Could they explain why this happens?
Author: Ali Mostafazadeh on 2022-01-06 [id 2071]
(in reply to Report 2 on 2022-01-03)
This referee has expressed his assessment by filling the online assessment form of SciPost and also attaching a PDF of his/her report whose content is slightly different. His/Her recommendation in the assessment form reads: ``My recommendation is that the paper should be revised significantly before acceptance in a more technical journal.'' while in the PDF (s)he writes: ``My recommendation is that the paper should be revised significantly before acceptance because it is difficult to appreciate many of the presented results." This is in sharp contrast with the opinion of the first referee who had no problem appreciating the significance of these results and recommended the publication of our paper by scoring its clarity as ``high.''
The main criticism of the second referee is that we have included too many definitions and equations without explanations, and that we should have instead provided graphical demonstration of our results. We believe that for a basic contribution to scattering theory, such as the one we report in this paper, the number of equations or lack of nice-looking graphs should have no baring on the significance of the results. The same criticism can also be made about the ground-breaking 1950 paper of Lippmann and Schwinger [Phys. Rev. 79, 469-480 (1950)] which includes 156 equations and no figures or appendices. We would definitely consider removing any of equations or their derivations that the referee finds unnecessary had (s)he specified their equation numbers. In a future revision we will consider moving some of the mathematical derivations to appendices, if this will ease the readability of the manuscript and does not hurt its coherence.
We agree that we could have offered more detailed explanation of the meaning of some of the equations, but we consider the present form of the manuscript sufficiently detailed. This view is shared by the first referee who stresses in his/her report that our presentation is absolutely self-contained and clear. We wish to draw attention to the fact that, by the very nature of the subject, one needs to have sufficient background in quantum mechanics to follow and appreciate this work. This does not however imply that it is a technical development beyond the reach of a general theoretical physicist. Indeed, every serious student of theoretical physics with a working knowledge of quantum mechanics (say at the level of Sakurai's QM) should be able to follow the arguments and repeat the calculations we present in our paper. Obviously, this requires certain amount of time and effort. After all it took us several months of intensive work to solve the multitude of problems we encountered in pursuing this line of research.
As far as the significance of our findings are concerned, we wish to recall the following points which we have underlined in the abstract, introduction, and the conclusion of the paper.
(i) This paper is the first to reveal the relevance of the notions of ``pseudo-Hermitian operator'' and ``exceptional point'' in the scattering theory of real potentials.
(ii) It is the first to offer a complete analytic solution of the scattering problem for a waveguide with a finite length when the source of the incident wave and the detectors are not placed at the ends of the waveguide but at spatial infinities.
(iii) It is the first to explain a concrete physical effect related to the presence of an exceptional point in waveguide system with no regions of gain or loss.
We believe that these are significant developments on their own sake, not to mention that they follow from what this referee qualifies as ``a novel scattering-matrix formulation.'' Our results are not restricted to a waveguide system but to a large class of physically relevant scattering problems whose treatment is beyond the standard methods of scattering theory. We are surprised how the referee could ignore these points, and conclude that the significance of our work is ``low."
Both version of the second referee's report includes 10 numbered more specific remarks/questions. The following are our response to these remarks.
Remark 1. The referee asks: ``Are the exceptional points that you find related to the exceptional points associated with total internal reflection?''
Our response to Remark 1: We are not aware of any reference on exceptional points for a waveguide that does not include gain or loss regions. Therefore, we do not understand what the referee means by ``the exceptional points associated with total internal reflection.''
Remark 2. The referee recommends that we write a sequel to our previous work on scattering theory where we ``can state the main equations one should use in order to construct the relevant operators in order to apply the method – rather than derive many results that were already published and present more than 100 equations. recollect the steps.''
Our response to Remark 2: The referee is under the impression that in this paper we ``derive many results that were already published.'' A quick examination of the references to our earlier works (Refs. 32 and 33) can easily show that this claim is not true. The present paper is the first in which we apply the ideas developed in Ref. 33 to potentials that decay to zero at spatial infinity only along some directions and blow-up along others. Standard methods of scattering theory (which are based on Lippmann-Schwinger equation) are not capable of dealing with such potentials. We emphasize that we have not derived any results in this paper that is already published by us or others. We did precisely what the referee suggests, i.e., recollected the necessary steps of our earlier work (Ref. 33) that we needed in dealing with potentials of the form (1). This corresponds to the content of the text that appears between Eqs. (4) -- (31), excluding the discussion of the reflection and transmission amplitudes and the S-matrix [given in Eqs. (15), (16), and (19) -- (21)] which we present for the first time. Our review of the earlier results occupies less than 15\% of the volume of our manuscript. The rest of the paper reports absolutely original developments. This includes the very definition of the effective Hamiltonian (33) which possesses the exceptional points. Therefore, we do not understand how the referee has reached the conclusion that we ``derive many results that were already published and present more than 100 equations.''
Remark 3. The referee writes ``I think it would be good to mention the advantages of the current formulation over previous scattering-matrix formulations of two-dimensional problems that require integration over momenta.''
Our response to Remark 3: We believe the referee means ``transfer-matrix formulation'' rather than ``scattering-matrix formulation.'' We have offered a brief discussion of the advantages of our approach and the earlier works on transfer matrix in two and three dimensions in Ref. 33. We did not include a more detailed discussion in the present paper, because we intended to keep it focused on the specific problems we consider. We also believe that such a comparison should be given in a comprehensive review article on the subject.
Remark 4. The referee asks: ``In Eq. 12, is $p_0$ defined anywhere?''
Our response to Remark 4: The answer to the referee's question is ``yes.'' It is given in Eq. (81). This is perhaps the only instance in our manuscript where we missed to give the definition of a symbol before or right after it appears. This is however not a major shortcoming, because from Eq. (12) it is almost obvious what $p_0$ should mean. In the revised manuscript we will give its definition also below Eq. (12).
Remark 5. The referee writes: ``The paper defines many quantities without explaining their physical meaning. To facilitate the reading, perhaps the authors can add explanations. For example, consider adding after the definition in Eq. (26) that it is an expansion of the identity operator in momentum space with weights w(p).''
Our response to Remark 5: Frankly, we do not subscribe to the assertion that every mathematical quantity entering the solution of a physics problem must be given a physical explanation or interpretation. What is necessary and often missing in the works of mathematical physicists is that they do not even give an explanation or interpretation of the final result of a mathematical analysis which aims to deal with a physics problem. This does not apply to our work, we use a formalism to solve a class of physically relevant scattering problems and present a clear interpretation of our final results, explaining the physical effect of the presence of an exceptional point on the behavior of scattering amplitude. It is true that offering physical explanation for intermediate mathematical results may be helpful. But in our opinion it is not absolutely necessary. Depending on the problem at hand, it can make the analysis of the problem too lengthy and divert the attention of the reader. For example, the referee suggests that we consider interpreting Eq.~(26), which reads
\[\widehat\varpi:=\varpi(\hat p)=
\int_{-\infty}^\infty dp\,\varpi(p)|p\rangle\langle p|,~~~~~~~~~~~~~~(26)\]
as ``an expansion of the identity operator in momentum space with weights w(p).'' This is actually not how we wish the reader to view this equation. Eq. (26) defines $\widehat\varpi$ as the value of the function $\varpi$ at the operator $\hat p$ and then reminds the reader how one defines functions of a Hermitian operator in terms of its spectral resolution. This is basic knowledge of undergraduate QM, which in our opinion needs no explanation or interpretation in a research paper on quantum scattering. For example, when we give the following formula for the evolution operator of a free particle of mass $m$ in one dimension,
\[\exp\left[-\frac{i(t-t_0)\hat p^2}{2m\hbar}\right]:=\int_{-\infty}^\infty dp\,
\exp\left(-\frac{i(t-t_0)p^2}{2m\hbar}\right)|p\rangle\langle p|, \]
we do not wish to interpret it as ``an expansion of the identity operator in momentum space with weights $\exp(-\frac{i(t-t_0)p^2}{2m\hbar})$.'' This is simply because this quantity is not the identity operator (except for $t=t_0$.) Before giving the above relation for the evolution operator, we need to give the definition of the functions of a Hermitian operator in terms of their spectral resolution. Then when we give this relation, the student will take it as a specific example of what a function of a Hermitian operator means. In our opinion, for a student who has exposure to basic operator theory of elementary QM, Eq.(26) will require no interpretation; it is just another example of what the function of a Hermitian operator means.
Remark 6. The referee asks the following questions: 6.a)``Is the number of real eigenvalues of $H_{\rm hat}$ (on page 8) equal to the number of bound states in the waveguide?'' 6b) ``Do the complex-conjugate pairs correspond to the continuum of unbounded solutions?'' 6.c) ``What is the physical significance of their imaginary components? (penetration depths? lifetimes?)''
Our response to Remark 6: As we have tried to make it clear from the beginning the effective Hamiltonian $\widehat{\mathbf{H}}(x)$ has been introduced as a tool that allows us to determine the transfer matrix of the system. From his/her questions, it seems that the referee confuses this effective Hamiltonian with the Hamiltonian, $\mathbf{p}^2+v(\hat x,\hat y)$, that defines the scattering problem through the Schr\"odinger equation~(2).
Our answer to the above questions are as follows: 6.a) For the waveguide system, the number of real eigenvalues of $\widehat{\mathbf{H}}(x)$ is given by Eq. (124). It is clear from this relation that this number depends on the incident wave number. Therefore, even for cases where the waveguide has no bound states (i.e., $\mathscr{V}_0\geq 0$) the effective Hamiltonian $\widehat{\mathbf{H}}(x))$ can have real eigenvalues. Therefore, the answer to 6.a) is ``No.'' 6.b) and 6.c) Again for the same reason the answer is ``No.'' The complex eigenvalues always exist irrespectively of whether the guide has a bound state or not. In the generic case the real eigenvalues are also present and these also contribute to the scattering phenomenon (behavior of scattering states.) We have provided a detailed discussion of the contribution of real and complex eigenvalues and the corresponding eigenvectors of $\widehat{\mathbf{H}}(x)$ to the scattering amplitudes of the waveguide in Sec.~5. See the paragraphs containing Eqs. (124) -- (145). The referee seems to have ignored this discussion and instead tried to adopt the standard interpretation of real and complex eigenvalues of complex Hamiltonians modeling infinite waveguides with active regions to gain a qualitative understanding of those of the effective Hamiltonian $\widehat{\mathbf{H}}(x)$. This is simply not warranted.
Remark 7. For the waveguide system of Sec. 5, the referee suggests us to plot ``the eigenvalues and eigenfunctions, showing that the spectrum contains exceptional points'' and ``the dependence of scattering amplitudes on the geometrical or material parameters.'' He also write: ``Any plot of this sort and its discussion could have helped me appreciate the findings.''
Our response to Remark 7: We can certainly generate and add such plots to the paper. In the original manuscript, we did not do so because we were not sure how helpful they would be. For example, the eigenvalues are given by Eq. (112). So they are either real or imaginary and plotting them means plotting a bunch of square root functions. The same is true of eigenfunctions which are given by Eq. (92). Therefore plotting them means plotting a bunch of sine functions.
Remark 8. The referee suggests that we state our results on the dependence of the transmission amplitude on the length of the waveguide at an exceptional point in the abstract and introduction and that we should provide a graphical demonstration of this result.
Our response to Remark 8: We can do this in the revised version but again we have reservations on the real impact of the plots of transmission amplitude, for we give explicit analytic formulas which includes simple functions. For example, see Eq. (138). If we wish to plot this quantity we should fix all the variables except the length $a$. To demonstrate what happens at an exceptional point we should confine our attention to the case where only the term $n=1$ is present in (138). Plotting the real and imaginary parts of this quantity as a function of $a$ when we are at the exceptional point will give horizontal lines and when we are aways from the exceptional point we get sums of a sine and cosine functions. In preparing the revised manuscript, we will generate these graphs and try to see if their inclusion can improve the presentation of our results.
Remark 9. The referee asks us to elaborate a comment we make in the second paragraph of conclusions, namely ``In particular, for an empty waveguide, we have shown that at the exceptional wavenumbers, where H develops an exceptional point, the transmitted wave includes a term that is independent of the length of the waveguide.'' (S)he then writes: ``Does this refer to the same point as my previous remark or is this a different result? Perhaps the authors could plot the contribution to the transmission that is independent and the one that depends on the length, and show how they relate to each other?''
Our response to Remark 9: This result follows from Eq. (138) when the sum includes more than one term with the first representing an exceptional point. In the revised manuscript we also provide a discussion of this point in the paragraph containing Eq. (139). Regarding plotting the length-dependent and length-independent contributions to the transmission amplitude, we will consider their inclusion in the revised manuscript after we generate them and make sure that they can be of help for some readers.
Remark 10. The referee writes: ``The authors say in the concluding paragraph that the addition of loss or gain would make the EPs disappear. Could they explain why this happens?''
Our response to Remark 10: We answer this question in the last paragraph of Sec. 5. As we discussed in great detail in Sec. 3, an exceptional point corresponds to the case where $w_n=0$. In the last paragraph of Sec. 5, we provide an argument showing that when the guide includes active or lossy material $w_n\neq 0$. Therefore, no exceptional point can exist.

---

## Round 2 · Referee Report · Anonymous · 2022-2-16

Strengths
1 - The paper presents a detailed, rigorous derivation of the authors' results
2 - It introduces a new approach to scattering theory, dealing with a specific class of 2D potentials
3 - It draws a connection between transport properties of systems without gain and loss and the non-Hermitian physics of exceptional points
Weaknesses
1 - Due to its nature, the paper is highly technical, which makes it sometimes difficult to follow
Report
The authors present a new approach to scattering theory, which applies to a range of two-dimensional problems for which the conventional scattering theory does not work. They use this framework to show a connection between transport properties in Hermitian systems and the exceptional points commonly associated with non-Hermitian systems, which would otherwise require gain and/or loss. The paper is detailed and self-contained, but it is highly technical, and it was difficult for me to follow the authors' derivation. However, I believe this is to be expected given the content of the paper and its results.
Given the above, my opinion is that this work does meet the acceptance criteria for publication in SciPost Physics (https://scipost.org/SciPostPhys/about). Specifically, it opens a new pathway in an existing research direction, with clear potential for multipronged follow-up work.
I have no changes to request, but I do have one question, mostly for the sake of satisfying my own curiosity. I would imagine that it is possible to associate the transfer matrix with an effective time-evolution operator governed by a non-Hermitian Hamiltonian also when doing the conventional scattering theory of Lippmann and Schwinger. Do you expect exceptional points to occur also in that case, or would you rather expect them to be a feature of the two-dimensional potentials you consider in this work?
Author: Ali Mostafazadeh on 2022-02-16 [id 2211]
(in reply to Report 2 on 2022-02-16)
This referee has also found our work meeting the acceptance criteria for publication in SciPost Physics and requested no changes. (S)he has asked us the following question: “I would imagine that it is possible to associate the transfer matrix with an effective time-evolution operator governed by a non-Hermitian Hamiltonian also when doing the conventional scattering theory of Lippmann and Schwinger. Do you expect exceptional points to occur also in that case, or would you rather expect them to be a feature of the two-dimensional potentials you consider in this work?”
In response to this question, we wish to point out that It is absolutely correct that we can use our approach to deal with the (short-range) potentials to which the standard scattering theory of Lippmann and Schwinger applies, and that the corresponding effective Hamiltonian will be non-Hermitian. Although it is not easy to make general statements about the structure and spectral properties of this Hamiltonian, we expect it to possess exceptional points whenever it has a point spectrum. The study of these exceptional points and their physical implications is an interesting topic for future research. Our work provides the first step towards exploring this topic.
Uwe Guenther on 2022-02-21 [id 2230]
The subsequent conceptual remark was part of an Invited Report finalized 4 hours after the Report deadline (end of the day of 16 Feb 2022) so that the SciPost reporting option was automatically closed and only comments were possible afterwards.
Remark:
On page 2, end of paragraph 2 the authors state:
"The present investigation differs from the earlier works on the physical aspects of exceptional points in that it deals with exceptional points arising in the treatment of a scattering problem for a real potential. In the context of their optical or acoustic realizations, these are exceptional points whose presence does not require active or lossy materials."
This statement can be embedded into a slightly broader conceptual context, having a closer glance at the Feshbach projection technique [r1,r2] as applied to derive from Hermitian setups (Hamiltonians with real potentials) effective non-Hermitian setups. This basic technique was used, e.g., to study the hidden subtleties of nuclear shell models [r3] and, in this way, it has been one of the starting points of the earlier investigations of I. Rotter [r4] leading later to her investigations into setups with exceptional points (EPs). Similarly, Feshbach-projected S-matrices have been used as theoretical background for the PRL series of the Darmstadt microwave experimental group around A. Richter [r5,11,r6] as well as in earlier slightly related mathematical studies [r7].
(Implicitly and roughly/conceptually spoken, the Naimark dilation approach of [r8] can be considered as a kind of "inverted Feshbach projection" technique in extending/dilating a given non-Hermitian setup into an associated Hermitian setup living in a Hilbert space of higher dimension.)
With regard to the present work, one can observe a strong conceptual analogy to Feshbach projections in the sense that an initial 2D-scattering setup with purely real potential is mapped (roughly spoken "projected") to an effective 1D evolution problem governed by the transfer-matrix-related non-Hermitian Hamiltonian. Clearly, the transfer matrix approach described in the present work (and developed in a series of earlier publications of the second author) is a deep novel finding whose importance is in no way diminished by very rough conceptual analogies to Feshbach projection techniques as known since the late 1950s. Rather, it can be identified as another important realization of a deep conceptual scheme which might be dubbed, e.g., a "generalized Feshbach-like dynamical projection" associating to a given higher-dimensional Hermitian setup an effective lower-dimensional non-Hermitian setup.
[r1] H. Feshbach, Ann. Phys. 5, 357 (1958).
[r2] H. Feshbach, Ann. Phys. 19, 287 (1962).
[r3] C. Mahaux and H.A. Weidenmueller, "Shell-model approach to nuclear reactions",
(North-Holland, Amsterdam, 1969).
[r4] I. Rotter, Rep. Prog. Phys. 54, 635 (1991).
[r5] B. Dietz et al., Phys. Rev. Lett. 98, 074103 (2007).
[r6] S. Bittner et al., Phys. Rev. Lett. 108, 024101 (2012).
[r7] S. Albeverio et al., J. Math. Phys. (N.Y.) 37, 4888 (1996).
[r8] U. Guenther and B.F. Samsonov, Phys. Rev. Lett. 101, 230404 (2008).

---

## Round 2 · Author Response

We hope that with the changes we have made in our manuscript, it is now suitable for publication in SciPost.

---

## Round 2 · List of Changes

We have added comments and new material to Sec. 2 (the line below Eq. 14), Sec. 3 (2nd line below the numbered list on page 8 and Footnote 3), and Sec. 5 (Fig. 3, which demonstrates the behavior of the eigenvalues and exceptional points of the effective Hamiltonian H, and its discussion given below Eq. 112, and Line 4 on page 22), and a new reference (Ref. 40). We have marked all changes made in the manuscript in red.

---

## Editorial Decision

published